JOURNAL OF
Neuroscience Research

# Laterodorsal tegmentum—ventral tegmental area projections encode positive reinforcement signals

Bárbara Coimbra[1,2] 🔗  |  Ana Verónica Domingues[1,2] 🔗  |  Carina Soares-Cunha[1,2] 🔗  |
Raquel Correia[1,2] 🔗  |  Luísa Pinto[1,2] 🔗  |  Nuno Sousa[1,2] 🔗  |  Ana João Rodrigues[1,2] 🔗

[1]Life and Health Sciences Research Institute (ICVS), School of Medicine, University of Minho, Braga, Portugal

[2]ICVS/3B's–PT Government Associate Laboratory, Braga/Guimarães, Portugal

**Correspondence**
Ana João Rodrigues, ICVS/School of Medicine, University of Minho, Campus de Gualtar, 4710-057 Braga, Portugal.
Email: ajrodrigues@med.uminho.pt

**Funding information**
CS-C and AJR have Scientific Employment Stimulus Contracts from Foundation for Science and Technology (FCT) (CEECIND/03887/2017; CEECIND/00922/2018). AVD has an FCT grant (SFRH/BD/147066/2019). The work was supported by a BIAL Foundation Grant (Bial 30/16); FCT project PTDC/MED-NEU/29071/2017 (REWSTRESS); part of the project received funding from "la Caixa" Foundation (ID 100010434), under the agreement LCF/PR/HR20/52400020; FCT - projects UIDB/50026/2020 and UIDP/50026/2020; NORTE-01-0145-FEDER-000013 and NORTE-01-0145-FEDER-000023.

## Abstract

The laterodorsal tegmentum (LDT) is a brainstem nucleus classically involved in REM sleep and attention, and that has recently been associated with reward-related behaviors, as it controls the activity of ventral tegmental area (VTA) dopaminergic neurons, modulating dopamine release in the nucleus accumbens. To further understand the role of LDT–VTA inputs in reinforcement, we optogenetically manipulated these inputs during different behavioral paradigms in male rats. We found that in a two-choice instrumental task, optical activation of LDT–VTA projections shifts and amplifies preference to the laser-paired reward in comparison to an otherwise equal reward; the opposite was observed with inhibition experiments. In a progressive ratio task, LDT–VTA activation boosts motivation, that is, enhances the willingness to work to get the reward associated with LDT–VTA stimulation; and the reverse occurs when inhibiting these inputs. Animals abolished preference if the reward was omitted, suggesting that LDT–VTA stimulation adds/decreases value to the stimulation-paired reward. In addition, we show that LDT–VTA optical activation induces robust preference in the conditioned and real-time place preference tests, while optical inhibition induces aversion. The behavioral findings are supported by electrophysiological recordings and c-fos immunofluorescence correlates in downstream target regions. In LDT–VTA ChR2 animals, we observed an increase in the recruitment of lateral VTA dopamine neurons and D1 neurons from nucleus accumbens core and shell; whereas in LDT–VTA NpHR animals, D2 neurons appear to be preferentially recruited. Collectively, these data show that the LDT–VTA inputs encode positive reinforcement signals and are important for different dimensions of reward-related behaviors.

**KEYWORDS**
LDT, motivation, neuronal circuits, optogenetics, reward

---

Edited by Bertrand Lambolez and Cristina Ghiani. Reviewed by Jerome Baufreton, Jacques Barik, and Jennifer Kaufling.

**Significance**

The laterodorsal tegmentum (LDT) inputs tightly control ventral tegmental area (VTA) dopaminergic activity, and thus have been associated with reward-related behaviors. This study shows that LDT–VTA projections' activation can increase reward value, enhance motivation, and drive positive reinforcement in male rats. Conversely, optical inactivation induces opposite behavioral outcomes. We further show that LDT–VTA input activation recruits VTA dopamine neurons and D1 neurons in the nucleus accumbens, while inhibition is associated with recruitment of D2 neurons. Together, these findings show that LDT–VTA projections are crucial for reward-related behaviors, and that its manipulation has a profound impact on behavioral output.

## 1 | INTRODUCTION

The laterodorsal tegmentum (LDT) has been classically associated with attention and REM sleep (Datta & Siwek, 1997; Dort et al., 2015; Redila et al., 2015; Thakkar et al., 1996), but recent evidence showed that this nucleus also plays a role in locomotion and in reward-related behaviors (Dautan, Souza, et al., 2016; Gut & Mena-Segovia, 2019; Lammel et al., 2012; Steidl & Veverka, 2015; Xiao et al., 2016).

The LDT contains populations of acetylcholine, glutamate, and GABA neurons (Luquin et al., 2018; Wang & Morales, 2009) that project to diverse areas of the brain, including the thalamus, ventral tegmental area (VTA), nucleus accumbens (NAc), among others (Cornwall et al., 1990; Holmstrand & Sesack, 2011; Luquin et al., 2018; Wang & Morales, 2009). This places the LDT in a privileged anatomical position to modulate diverse circuits in the brain, including the reward circuit. For long it is known that the LDT provides a regulatory input to the VTA (Lammel et al., 2012; Oakman et al., 1995; Watabe-Uchida et al., 2012; Woolf & Butcher, 1986). Specifically, the LDT provides asymmetric (excitatory type) inputs to VTA dopaminergic neurons that preferentially innervate the NAc (Omelchenko & Sesack, 2005). Additionally, LDT–VTA cholinergic terminals were found to synapse on VTA dopamine neurons that innervate the NAc (Omelchenko & Sesack, 2006). It has been proposed a divergent LDT influence on mesoaccumbens neurons that appears to excite dopaminergic cells and inhibit GABA neurons of the VTA (Omelchenko & Sesack, 2005, 2006). Indeed, previous reports have shown that electrical stimulation of the LDT increases NAc dopamine levels by activating VTA dopaminergic cells through both glutamatergic and cholinergic receptors (Forster & Blaha, 2000; Forster et al., 2002). Additionally, LDT activity is essential for VTA dopaminergic burst firing (Lodge & Grace, 2006), which is considered to be the functionally relevant signal that encodes reward or indicates incentive salience/motivation (Berridge & Robinson, 1998; Cooper, 2002; Grace & Bunney, 1984; Schultz, 1998). More recently, it has been shown that the LDT also sends direct projections to the NAc (Coimbra et al., 2019; Dautan et al., 2014; Dautan, Hacıoğlu Bay, et al., 2016), further supporting the importance of this brain region for the reward circuitry modulation.

Since LDT neurons are involved in the fine tuning of the VTA dopaminergic activity, it is becoming increasingly evident that this region plays an important role in reinforcement. Indeed, optogenetic excitation of LDT–VTA cells results in the acquisition of conditioned place preference (CPP) in rodents (Dautan, Souza, et al., 2016; Lammel et al., 2012; Xiao et al., 2016) and reinforces lever pressing in rats (Coimbra et al., 2017; Steidl, O'Sullivan, et al., 2017; Steidl & Veverka, 2015), suggesting that LDT–VTA inputs convey positive/rewarding signals. Less is known about the specific role of these projections in motivation and in reward value.

In this work we provide evidence about the role of LDT–VTA inputs in different dimensions of reinforcement, by optogenetically activating and inhibiting these inputs in a wide range of behavioral tests. Besides confirming previous studies of LDT–VTA involvement in inducing place preference and increasing operant behavior, we provide novel evidence about the role of LDT–VTA inputs in motivational drive and in value encoding.

## 2 | MATERIAL AND METHODS

### 2.1 | Animals and treatments

Male Wistar Han rats were individually housed under standard laboratory conditions (light/dark cycle of 12 hr; 22°C); food and water *ad libitum*, with enrichment materials in each cage (cardboard tubes, nesting materials). At the start of the experiments, 2–3-month-old males were used for electrophysiological and behavioral experiments. A limitation of this study is that we did not use females, however, it is important to refer that previous work from our group showed no significant differences in behavioral performance in the two choice task or progressive ratio in both sexes (data not shown).

Two different experiments were performed: in one group of animals we assessed electrophysiological evoked activity in the VTA by LDT terminal activation ($n_{ChR2} = 9$; $n_{NpHR} = 5$); and a different group of animals was used for behavioral experiments (at the start of experiment: $n_{YFP} = 10$, $n_{ChR2} = 14$; $n_{YFP} = 10$, $n_{NpHR} = 14$). All manipulations were conducted in accordance with European Regulations (European Union Directive 2010/63/EU). Animal facilities and the people directly involved in animal experiments were certified by the Portuguese regulatory entity—Direção Geral de Alimentação e Veterinária (DGAV). All experimental procedures are authorized by DGAV under project #23432 (2013) and #19074 (2016).

### 2.2 | Constructs and virus preparation

AAV5–EF1a–WGA–Cre–mCherry, AAV5–EF1a–DIO–hChR2–YFP, AAV5–EF1a–DIO–NpHR3.0–YFP, and AAV5–EF1a–DIO–YFP were obtained directly from the Gene Therapy Center Vector Core (UNC)

center. AAV5 vector titers were $2.1$–$6.6 \times 10^{12}$ virus molecules/ml as determined by dot blot.

## 2.3 | Surgery and cannula implantation

Rats designated for behavioral experiments were anesthetized with 75 mg/kg ketamine (Imalgene, Merial) plus 0.5 mg/kg medetomidine (Dorbene, Cymedica). 0.5 µl of AAV5–EF1a–WGA–Cre–mCherry and AAV5–EF1a–DIO–hChR2–YFP were unilaterally injected into the VTA (coordinates from bregma, according to Paxinos and Watson (2007): −5.4 mm anteroposterior, +0.6 mm mediolateral, and −7.8 mm dorsoventral) and LDT (coordinates from bregma: −8.5 mm anteroposterior, +0.9 mm mediolateral, and −6.5 mm dorsoventral), respectively (ChR2 group) or 0.5 µl of AAV5–EF1a–WGA–Cre–mCherry into the VTA and AAV5–EF1a–DIO–NpHR3.0–YFP in the LDT (NpHR group). Another group of animals was injected with 0.5 µl of AAV5–EF1a–WGA–Cre–mCherry in the VTA, and in the LDT, with 0.5 µl AAV5–EF1a–DIO–YFP (YFP groups). Rats were then implanted with an optic fiber (200 µm core fiber optic; Thorlabs) with 2.5 mm stainless steel ferrule (Thorlabs) using the injection coordinates for the VTA (with the exception of dorsoventral: −7.7 mm) that were secured to the skull using 2.4 mm screws (Bilaney, Germany) and dental cement (C&B kit, Sun Medical). Rats were removed from the stereotaxic frame and sutured. Anesthesia was reverted by administration of atipamezole (1 mg/kg). After surgery animals were given anti-inflammatory (Carprofeno, 5 mg/kg) for 1 day, analgesic (butorphanol, 5 mg/kg) for 3 days, and were let to fully recover for 30 days before initiation of behavior, to allow viral expression. Optic fiber placement was confirmed for all animals after behavioral experiments (Figure S1a, at the start of experiment: $n_{YFP} = 10$, $n_{ChR2} = 14$; $n_{YFP} = 10$, $n_{NpHR} = 14$). Animals that were assigned for electrophysiological experiments were not implanted with an optic fiber.

## 2.4 | Behavior

Experimental design with groups and number of animals is depicted in Figure S2. Number of animals for behavioral experiments varied, considering that animals who lose fiber implants were removed from the experiment. Number of animals for behavioral experiments are as follows: real-time place preference (RTPP) and CPP—$n_{YFP} = 10$; $n_{ChR2} = 14$; $n_{YFP} = 10$; $n_{NpHR} = 14$; Operant behavior (Two-choice, PR, Extinction)—$n_{YFP} = 10$; $n_{ChR2} = 11$; $n_{YFP} = 10$; $n_{NpHR} = 11$. Experiments comprising YFP and ChR2 groups were replicated three times. Experiments comprising YFP and NpHR groups were performed once, considering the representative number of animals.

## 2.5 | Subjects and apparatus

Rats were placed and maintained on food restriction (≈7 g/day of standard laboratory chow) to maintain 90% free-feeding weight. Behavioral sessions were performed in operant chambers (Med Associates)

containing a central magazine that provided access to 45 mg standard food pellets (F0021, Dustless Precision Pellets, Bio-Serve), two retractable levers located on each side of the magazine with cue lights above them. A 2.8 W, 100 mA house light positioned at the top center of the wall opposite to the magazine provided illumination. A computer equipped with Med-PC software (Med Associates) controlled the equipment and recorded the data.

## 2.6 | Two-choice schedule of reinforcement

During instrumental training, rats are presented two illuminated levers, one on either side of the magazine. Presses on one lever (Laser + pellet delivery (stim+ lever)) leads to instrumental delivery of a pellet plus 4 s blue (473 nm—80 10 ms pulses at 20 Hz) or yellow (589 nm—constant light) laser stimulation at 10 mW, paired with a 4 s auditory cue. In contrast, pressing the other lever (stim− lever) delivered a single pellet paired with another 4 s auditory cue, but with no laser stimulation. For both levers, presses during the 4 s after pellet delivery have no further consequence. After 2 days of habituation, each daily session begins with a single lever presented alone to allow opportunity to earn its associated reward (either stim+ or stim−), after which the lever is retracted. Then, the alternative lever is presented by itself to allow opportunity to earn the other reward, to ensure that the rat sampled both reward outcomes. Finally, both levers together are extended for the remainder of the session (30 min total), allowing the rat to freely choose between the two levers and to earn respective rewards in any ratio (FR1, FR4, RR4, and RR6).

## 2.7 | Progressive ratio

The progressive ratio (PR) test was performed for either the stim+ or stim− lever in separate sessions, repeating the same conditions as described above: after lever press requirement achieved, for stim+ lever, pellet delivery was coupled to optical stimulation (4 s): blue laser, 473 nm—80 10 ms pulses at 20 Hz; or yellow laser, 589 nm—constant light at 10 mW. For stim− lever, reward consisted of a single pellet with no stimulation. The order of test conditions is counterbalanced across animals and repeated for each animal with the other lever. The number of presses required to produce the next reward delivery increases after each reward, according to an exponential progression (PR schedule: 1, 2, 4, 6, 9, 12, 15, 20, 25, 32, 40, 50, 62, 77, 95, 118, 145, 178, 219, 268,...) derived from the formula PR = [5e(reward number * 0.2)] – 5 and rounded to the nearest integer. To determine whether any preference in responding is the result of increased workload, animals are given a FR1 session after PR, identical to the initial day of training.

## 2.8 | Extinction of food in the two-choice schedule of reinforcement task

To conversely assess whether laser stimulation alone can maintain responding when the reward is discontinued, rats are given the

opportunity to press the same levers but without pellet (pellet extinction). Each completed trial (RR4) on the stim+ lever results in the delivery of laser stimulation and the previously paired auditory cue but no pellet delivery. Each completed trial on the other lever (stim−—previously pellet alone) resulted in the delivery of its auditory cue.

## 2.9 | Conditioned place preference

The CPP apparatus consisted of two compartments with different patterns, separated by a neutral area (Med Associates): a left chamber measuring 27.5 cm × 21 cm with black walls and a grid metal floor; a center chamber measuring 15.5 cm × 21 cm with gray walls and gray plastic floor; and a right chamber measuring 27.5 cm × 21 cm with white walls and a mesh metal floor. Rat location within the apparatus during each preference test was monitored using a computerized photo-beam system (Med Associates). Briefly, on day 1, individual rats were placed in the center chamber and allowed to freely explore the entire apparatus for 15 min (pre-test). On day 2, rats were confined to one of the side chambers for 30 min and paired with optical stimulation—ON side; in the second session, rats were confined to the other side chamber for 30 min with no stimulation—OFF side. Optical stimulation consisted of 80 pulses of 10 ms at 20 Hz, every 15 s for blue light and 4 s of constant light at 10 mW, every 15 s for yellow light. Conditioning sessions were counterbalanced between animals. On day 3, rats were allowed to freely explore the entire apparatus for 15 min (test day). Results are expressed as the ratio of preference in the ON chamber and total time spent on each side of the apparatus.

## 2.10 | Real-time place preference

The RTPP test was performed in a custom-made black plastic arena (60 × 60 × 40 cm) comprised by two indistinguishable chambers for 15 min. One chamber was paired with either blue light stimulation of 10 ms pulses at 20 Hz or constant yellow light stimulation, during the entire period that the animal stayed in the stimulus-paired side. The choice of paired chamber was counterbalanced across rats. Animals were placed in the no-stimulation chamber at the start of the session and light stimulation started at every entry into the paired chamber. Animal activity was recorded using a video camera and time spent in each chamber was manually assessed. Results are presented as percentage of time spent in each chamber.

## 2.11 | In vivo single cell electrophysiological recordings

Experimental design with groups and number of animals is depicted in Figure S2a.

Four weeks after viral injection, animals were submitted to a stereotaxic surgery for the placement of the optic fiber coupled with tungsten recording electrode. Animals were anesthetized with urethane (1.44 g/kg, Sigma). The total dose was administered in three separate intra-peritoneal injections, 15 min apart. Body temperature was maintained at ~37°C with a homeothermic heat pad system (DC temperature controller, FHC, ME, USA). Adequate anesthesia was confirmed by observation of general muscle tone, by assessing withdrawal responses to noxious pinching and by whisker movement.

Recording electrode coupled with a fiber optic patch cable (Thorlabs) was placed in the following coordinates: VTA: −5.4 from bregma, 0.6 lateral from midline, −7.5 to −8.2 ventral to brain surface. A reference electrode was fixed in the skull, in contact with the dura. We recorded nine animals for the ChR2 group and five for the NpHR group, averaging nine cells per animal.

Extracellular neural activity from the VTA was recorded using a tungsten recording electrode (3–7 MW at 1 kHz), that was lowered in increments of 100 μm, from −7.5 to −8.2. Recordings were amplified and filtered by the Neurolog amplifier (NL900D, Digitimer Ltd, UK) (low-pass filter at 500 Hz and high-pass filter at 5 kHz).

Spontaneous activity of single neurons was recorded to establish baseline firing rate for at least 60 s, as averaged number of spikes that occur in a 60 s period. The DPSS 473 nm and 589 nm laser system (CNI), controlled by a stimulator (Master-8, AMPI), was used for intracranial light delivery and fiber optic output was pre-calibrated to 10 mW. Optical stimulation was performed for each detected single neuron and consisted of:

- Optical activation: ChR2 group, 80 pulses of 10 ms at 20 Hz of blue laser;
- Optical inhibition: NpHR group, 4 s of constant yellow light;

Spikes of a single neuron were discriminated, and data sampling was performed using a CED micro 1401 interface and SPIKE 2 software (Cambridge Electronic Design, Cambridge, UK).

Firing rate was established for the baseline, stimulation period, and post-stimulation period (60 s after the end of stimulation).

Neurons showing a firing rate increase or decrease by more than 20% from the mean frequency of the baseline period were considered as responsive. A criterion of change of firing rate 20% above or below average activity of the baseline was used, as previously reported by others and us (Benazzouz et al., 2000; Coimbra et al., 2017, 2019; Soares-Cunha et al., 2016, 2020). A heatmap of neuronal response (in percentage) was generated (Figure 1j,m), comprising the following periods: 2 s pre-stimulus (baseline), stimulus and 2 s post-stimulus activity, using 50 ms time bins. We classified single units in the VTA into three separate groups of putative neurons: putative dopamine (DA), putative GABA, and "other" neurons. This classification was based on firing rate and waveform duration (calculated from average spike waveform) (Totah et al., 2013; Ungless & Grace, 2012; Ungless et al., 2004). Cells presenting a firing rate <10.0 Hz and a duration of >1.5 ms were considered putative DAergic (pDAergic) neurons. If

the firing rate was >10.0 Hz and waveform duration <1.5 ms, cells were assigned to putative GABAergic (pGABAergic) neuron group. Other single units were excluded from analysis ($n = 7$ cells). This group likely contains units from both DA and GABA groups. At the end of each electrophysiological experiment, all brains were collected and processed to identify recording region.

## 2.12 | Immunofluorescence

Animals were anesthetized with pentobarbital (Eutasil, Lisbon, Portugal) and transcardially perfused with 0.9% saline followed by 4% paraformaldehyde. Brains were removed and sectioned sagittally at a thickness of 50 μm, on a vibrating microtome (VT1000S,

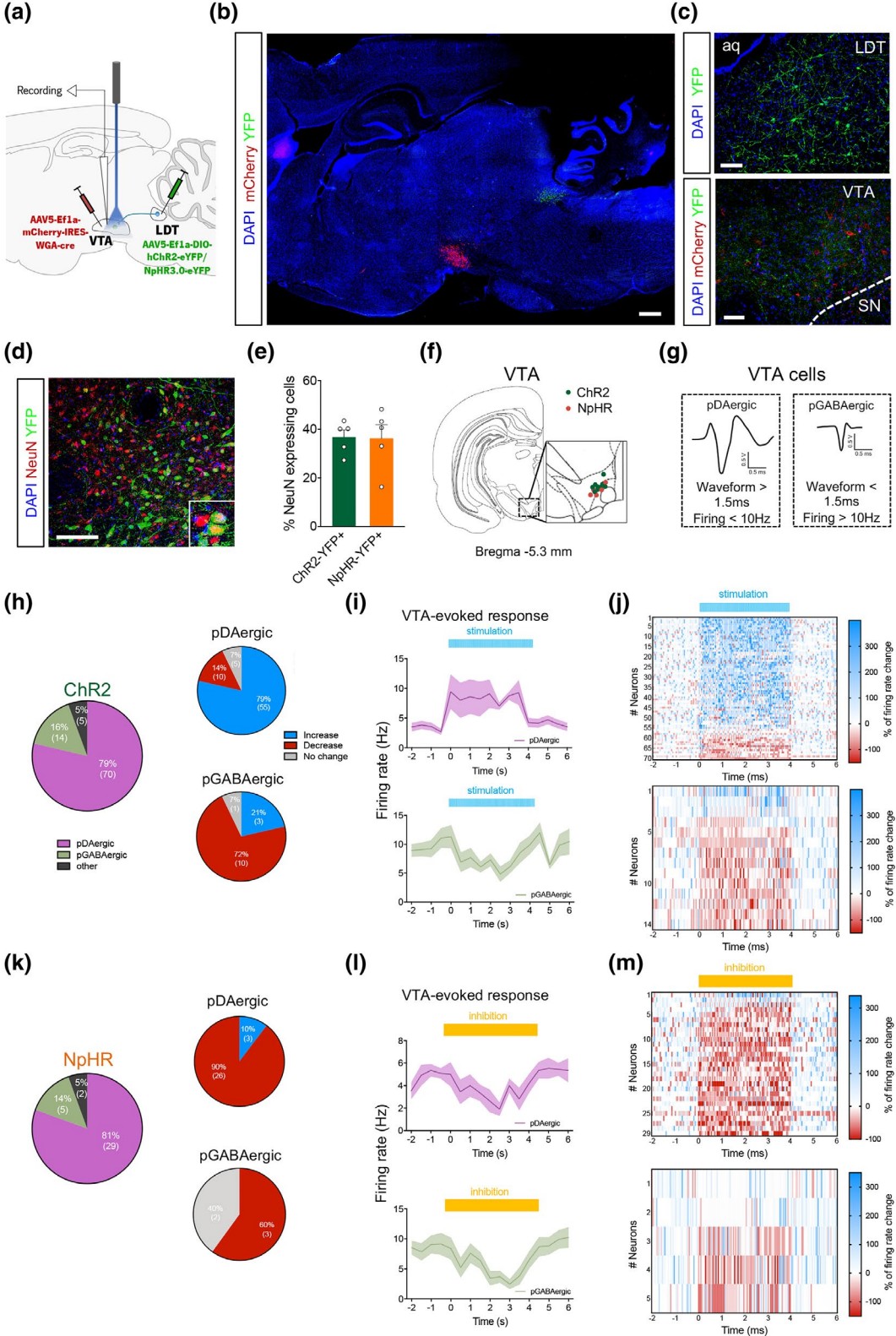

**FIGURE 1** Optogenetic modulation of LDT–VTA projections alters VTA neuronal activity. (a) Strategy used for LDT–VTA projection optogenetic stimulation and electrophysiological recordings in the VTA. (b) Representative immunofluorescence showing eYFP (green) expression in the LDT and mCherry (red) in the VTA; scale bar = 1 mm. (c) Representative immunofluorescence showing YFP staining in the LDT and in terminals in the VTA; scale bar = 100 μm. (d) Representative immunofluorescence showing YFP and NeuN staining in the LDT; scale bar = 100 μm. NeuN immunoreactivity identifies all cells in the LDT. (e) Summary graph showing the proportion of LDT cells (identified by NeuN-red) that express either opsin, ChR2 or NpHR (identified by GFP-green). ChR2 (*n* = 5) and NpHR (*n* = 5) animals. (f) Electrode placement for cell recording in the VTA for ChR2 (*n* = 9) and NpHR (*n* = 5) animals. (g) VTA neurons were separated into putative dopaminergic cells (pDAergic) and putative GABAergic cells (pGABAergic). (h) Majority of recorded cell in the ChR2 group are dopaminergic neurons. The majority of dopaminergic cells neurons significantly increase firing rate, whereas the opposite is observed in GABAergic cells, in response to optical stimulation (80 pulses of 10 ms at 20 Hz) of LDT terminals (*n* = 89 neurons/9 rats). (i) Temporal activity (0.5 s bins) of VTA pDAergic (upper panel) and pGABAergic (bottom panel) cells in response to LDT optical stimulation. Full line trace represents mean frequency of recorded cells and *SEM* as error is represented as shading. (j) Heatmap representation of percentages of pDAergic (upper panel) and pGABAergic (bottom panel) cell responses in the VTA upon activation of LDT terminals. (k) The majority of recorded cell in the NpHR group is dopaminergic neurons. Most of pDAergic and pGABAergic cells neurons significantly decrease firing rate, in response to optical inhibition (4 s of continuous yellow laser) of LDT terminals (*n* = 34 neurons/5 rats). (l) Temporal activity (0.5 s bins) of VTA pDAergic (upper panel) and pGABAergic (bottom panel) cells in response to LDT terminal inhibition. Full line trace represents mean of recorded cells and *SEM* as error is represented in the shading. (m) Heatmap representation of percentages of pDAergic (upper panel) and pGABAergic (bottom panel) cell responses in the VTA upon inhibition of LDT terminals. Bars represent mean and error bars denote *SEM*. *$p < 0.05$

Leica, Germany). Sections were incubated with the primary antibody chicken anti-mCherry (1:1,000, HBT008-100, HenBiotech) and rabbit anti-GFP (1:1,000, ab290, Abcam), followed by appropriate secondary fluorescent antibodies (1:1,000, anti-chicken Alexa Fluor 594, A-11042, anti-rabbit Alexa Fluor 488, A-21206, Invitrogen).

For c-fos experiments, animals were anesthetized with pentobarbital (Eutasil) 90 min after initiation of the PR test (vide experimental design in Figure S2a), and transcardially perfused as described above. Coronal sections were incubated with one of the following primary antibodies: mouse anti-TH (1:1,000, MAB318, Millipore); rabbit anti-c-fos (1:1,000, Ab-5, Millipore), goat anti-ChAT (1:750, AB144P, Millipore), sheep anti-ChAT (1:750, ab18736, Abcam), mouse anti-dopamine D1 receptor (1:300, sc-33660, Santa Cruz Biotechnology), mouse anti-dopamine D2 receptor (1:400, sc-5303, Santa Cruz Biotechnology), followed by appropriate secondary fluorescent antibodies (1:1,000) (anti-goat Alexa Fluor 594, A-11058, anti-mouse Alexa Fluor 594, A32744, anti-sheep Alexa Fluor 647, A-21448, anti-rabbit Alexa Fluor 488, A-21206, all from Invitrogen). Sections were stained with 4′,6-diamidino-2-phenylindole (DAPI; 1 mg/ml, D1306, Invitrogen) and mounted using mounting media (Permafluor, Invitrogen).

Slices for selected brain regions were the following (according to Paxinos coordinates (Paxinos & Watson, 2007)): LDT—from bregma −8.4 to −9.1; VTA—from bregma −5.2 to −6.3; and NAc—from bregma +2.8 to +0.8.

Image acquisition was performed by confocal microscopy (Olympus FV1000, Olympus) under 20× magnification. Confocal images were analyzed using ImageJ. Sections were labeled and areas delimited relative to bregma using landmarks and neuro-anatomical nomenclature as described in Paxinos and Watson (2007). Positive cells within the brain regions of interest were manually analyzed in a blind manner, the same five sections per region per animal were considered (animals that performed task for stim+ lever: $n_{YFP}$ = 5; $n_{ChR2}$ = 5; $n_{YFP}$ = 5; $n_{NpHR}$ = 5; animals that performed task for stim− lever: $n_{YFP}$ = 5; $n_{ChR2}$ = 5; $n_{YFP}$ = 5; $n_{NpHR}$ = 5). Estimation of cell density of c-fos positive cells and double positive cells—cell number divided by the corresponding areas—was obtained for each region.

Medial and lateral VTA subregions were selected according to the anatomical location of distinct dopaminergic sub-populations, as previously described (Beier et al., 2015; Lammel et al., 2008, 2011; Yang et al., 2018). The medial VTA was considered as the region comprising the paranigral nucleus and interfascicular nucleus and the lateral VTA was defined as the lateral parabrachial pigmented nucleus and the medial lemniscus adjacent to the substantia nigra. In addition, we divided NAc into subregions, core, and shell (Aragona et al., 2008; Bassareo et al., 2002; Dreyer et al., 2016).

To quantify the percentage of LDT neurons transfected with ChR2 or NpHR, we used slices containing the LDT of animals from electrophysiological experiments, from the coordinates mentioned above ($n_{ChR2}$ = 5 animals; $n_{NpHR}$ = 5 animals). Coronal sections were incubated overnight with the following primary antibodies: mouse anti-NeuN (1:750, MAB377, Millipore) and rabbit anti-GFP (1:1,000, ab290, Abcam), followed by appropriate secondary fluorescent antibodies (1:1,000, anti-mouse Alexa Fluor 594, A32744, anti-rabbit Alexa Fluor 488, A-21206, Invitrogen). Image acquisition was performed as mentioned, by confocal microscopy (Olympus FV3000, Olympus) under 20× magnification and analyzed using ImageJ. Sections were labeled and areas delimited in the same manner as above. Positive cells within the LDT were manually analyzed in a blind manner and the same five sections per animal were considered. Quantification of proportion of LDT neurons (as assessed by NeuN-positive cells) that expressed ChR2 or NpHR (GFP-positive cells) was obtained by dividing the number of GFP-positive cells by the number of NeuN-positive cells in the LDT.

## 2.13 | Statistical analysis

Statistical analysis was performed in GraphPad Prism 8.0 (GraphPad Software, Inc., La Jolla, CA, USA) and SPSS Statistics v24.0 (IBM corp, USA). Parametric tests were used whenever Shapiro–Wilk normality test SW >0.05.

Statistical analysis between two groups was made using Student's *t* test. Non-parametric analysis (Mann–Whitney test) was used when

normality of data was not assumed. Repeated measures two-way analysis of variance (ANOVA) was used to compare groups versus sessions, groups versus positive cell density. Bonferroni's post hoc multiple comparison test was used for group difference determination. Report of statistical values is described along with results.

We compared neuronal firing rate distributions (baseline vs. stimulation) in the VTA (dopamine and GABA identified cells) using the two-sample Kolmogorov–Smirnov test (0.05 s bins spanning from 2 s before laser stimulation, during laser stimulation of 4 s, through 2 s after laser stimulation).

Pearson's correlation was used to examine the relationship between recruited c-fos+ cells and breakpoint levels reached in the progressive ratio task and between recruited c-fos+ cells in VTA and NAc subregions. Results are presented as scatterplots distribution and mean $\pm$ SEM. Statistical significance was accepted for $p < 0.05$.

## 3 | RESULTS

### 3.1 | Optogenetic modulation of LDT terminals changes VTA firing rate

We used a combined viral approach to specifically target LDT direct inputs to the VTA. We injected an adeno-associated virus (AAV5) containing a WGA–Cre fusion construct (AAV–EF1a–DIO–WGA–Cre–mCherry) in the VTA, and a vector encoding a cre-dependent ChR2 (optical excitation) or NpHR (optical inhibition) in the LDT (AAV–EF1a–DIO–hChR2–eYFP – ChR2 group; AAV–EF1a–DIO–NpHR–eYFP – NpHR group). Control animals were injected with AAV–EF1a–DIO–eYFP in the LDT (YFP groups). The WGA–Cre fusion protein is retrogradely transported (Gradinaru et al., 2010; Xu & Südhof, 2013), inducing the expression of cre-dependent ChR2- or NpHR–YFP only in LDT neurons that directly project to the VTA (Figure 1a,b). We were able to observe YFP staining throughout soma and dendrites of LDT neurons and in terminals located in the VTA, next to mCherry-positive cells (Figure 1c). In order to attest for the efficacy of the viral approach, we quantified the proportion of LDT neurons that expressed YFP (Figure 1d,e; percentage of cells expressing NeuN and YFP). 36.7% of NeuN-positive LDT neurons expressed ChR2–YFP and 36.3% expressed NpHR–YFP, after viral injections into the VTA and LDT.

To evaluate the functionality of the approach, we performed single cell electrophysiological recordings in anesthetized rats (Figure 1f–m), while optically stimulating LDT terminals in the VTA as previously described (Coimbra et al., 2017). Cells in the VTA were divided according to waveform duration and firing rate (Figure 1g) (Coimbra et al., 2017; Totah et al., 2013; Ungless & Grace, 2012; Ungless et al., 2004). Activation of LDT terminals (80 10 ms-light-pulse trains delivered at 20 Hz) evoked excitatory responses in 79% (55 of 70 cells) of VTA putative dopaminergic (pDAergic) neurons and inhibitory responses in 71% (10 of 14 cells) of VTA putative GABAergic (pGABAergic) recorded

neurons (Figure 1h; n = 89 total recorded cells; n = 9 rats). Optical activation induced a significant increase in the firing rate of VTA pDAergic neurons in comparison to baseline (Figure 1i, Kolmogorov–Smirnov test, two-tailed, D = 1.00, p = 0.0028). Firing rate of pGABAergic neurons decreased with activation of LDT terminals in comparison to baseline (Figure 1i, Kolmogorov–Smirnov test, two-tailed, D = 1.00, p = 0.0014). A heatmap of firing rates of pDAergic cells showed that majority of cells increased firing rate during optical stimulation whereas pGABAergic displayed a decrease (Figure 1j). The percentage of VTA cells that responded to LDT manipulation was quite high as described in other studies (Lammel et al., 2011; Lodge & Grace, 2006), although others showed a reduced percentage of responsive neurons (Dautan, Souza, et al., 2016; Fernandez et al., 2018; Xiao et al., 2016).

Regarding optical inhibition experiments, we observed that 4 s stimulation of LDT terminals in the VTA reduced activity in 89.7% (26 of 29 cells) of VTA pDAergic neurons and in 60% (3 of 5 cells) of VTA pGABAergic recorded neurons (Figure 1k; n = 34 total recorded cells; n = 5 rats). Analysis of distribution showed that optical inhibition of LDT–VTA projections decreased VTA pDAergic and pGABAergic neurons firing rate in comparison to baseline (Figure 1l, VTA pDAergic: Kolmogorov–Smirnov test, two-tailed, D = 0.7778, p = 0.0364; VTA pGABAergic: Kolmogorov–Smirnov test, two-tailed, D = 0.8889, p = 0.0140). A heatmap of firing rates of pDAergic cells showed that majority of cells decreased firing rate during optical stimulation (both pDAergic and pGABAergic), but returned to baseline activity after (Figure 1m).

### 3.2 | Optogenetic activation of LDT–VTA inputs enhances preference for a laser-paired reward

To test the impact of LDT–VTA manipulation on behavior, we used the previously described viral approach (Figure 2a) and unilaterally activated these inputs in freely moving rats during a two-choice instrumental task (Figure 2b). Animals were trained to press two levers to get food pellets; one of the levers was arbitrarily selected to deliver the pellet with simultaneous LDT–VTA optogenetic stimulation (stim+; blue laser: 80 10 ms pulses at 20 Hz), whereas pressing the other lever delivered only the pellet (stim−) (Figure 2b). The effort to get the pellet was increased until a random ratio of six lever presses per reward.

ChR2 rats progressively discriminate and prefer the stim+ lever throughout the acquisition stage (Figure 2c; RM two-way ANOVA, session: $F_{(3\cdot524, 133.9)} = 101.3$, $p < 0.0001$; group: $F_{(3, 38)} = 39.73$, $p < 0.0001$), without altering lever pressing performance, as the total number of lever presses was similar between ChR2 and YFP animals (Figure S3a). At the end of the acquisition stage, ChR2 animals exhibited a 4.6:1 ratio of preference for the stim+ lever in comparison to stim− lever (Bonferroni post hoc t test, $t_{(12.45)} = 10.98$, $p < 0.001$). YFP animals had no preference for either lever.

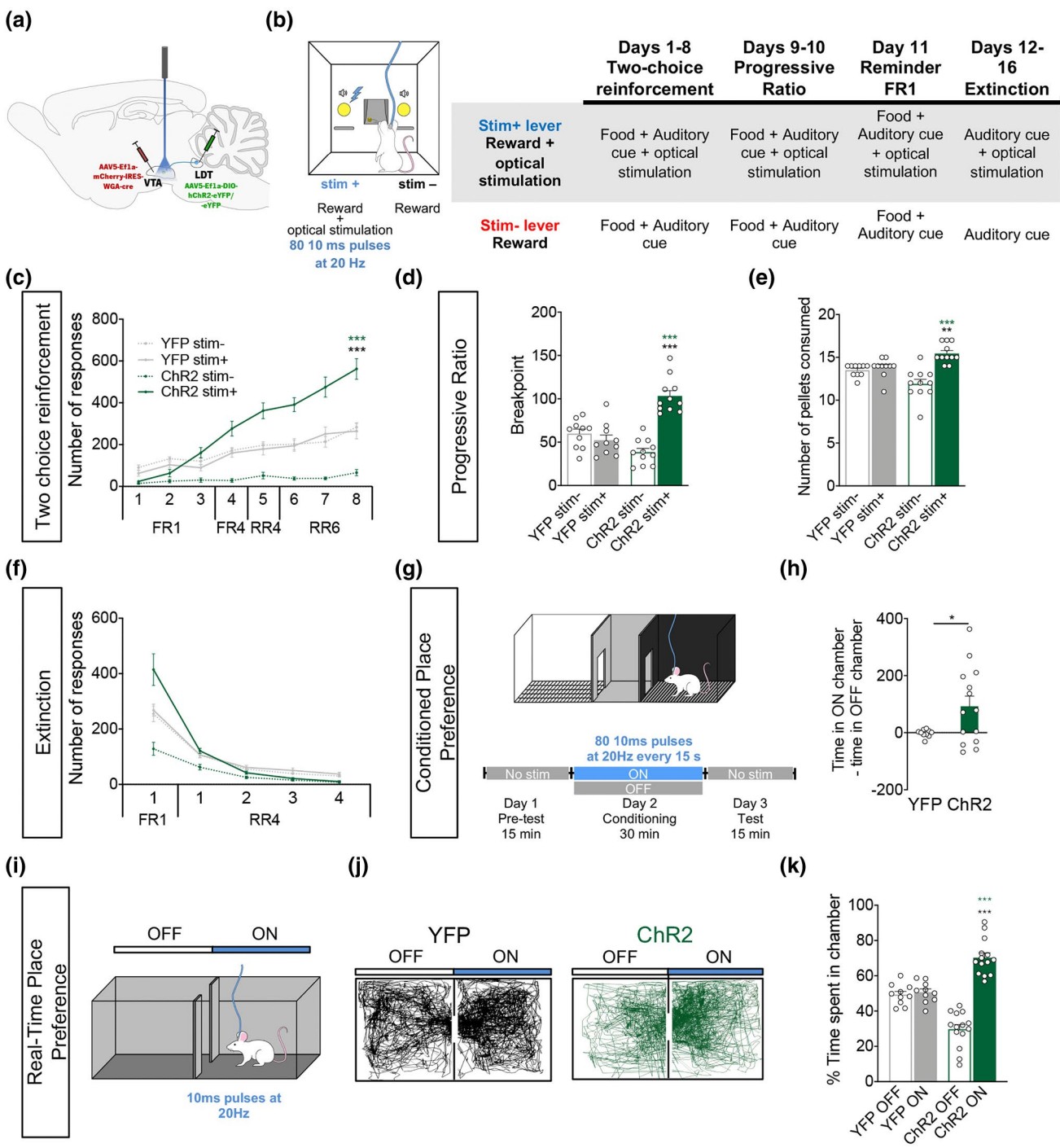

**FIGURE 2** Optogenetic activation of LDT terminals in the VTA increases motivation. (a) Strategy used for LDT–VTA projection optogenetic stimulation during behavior. (b) Schematic representation of the two-choice task. Pressing stim− lever yields one food pellet and pressing stim+ lever delivers one pellet + optical stimulation of LDT-VTA inputs (80 10 ms pulses at 20 Hz). (c) Time-course representation of the responses in ChR2 ($n = 11$) and YFP ($n = 10$) rats. Optogenetic activation of LDT–VTA terminals focuses responses for the lever associated with the laser-paired reward (stim+) over an otherwise equivalent food reward (stim−) in ChR2 animals, but not in control YFP group. (d) Rats were subjected to two PR sessions, one for each lever: in one session, animals are tested for the stim+ lever, and in the other session animals are tested for the stim− lever. We observe an increase in the breakpoint for stim+ lever in ChR2 animals, indicative of enhanced motivation. (e) Total number of rewards earned during progressive ratio of ChR2 and YFP rats. ChR2 animals worked to earn more rewards for the stim+ than the stim− lever or YFP animals. (f) In pellet extinction conditions, both groups decrease responses for both levers. (g) CPP and (i) RTPP paradigms, in which one chamber is associated with laser stimulation (ON side). (h) Difference of total time spent in the OFF and ON sides in YFP ($n = 10$) and ChR2 ($n = 14$) groups. (j) Representative tracks for a ChR2 and a YFP animal during the RTPP. (k) Percentage of time spent on the ON and OFF sides, showing preference for the side associated with stimulation. Error bars denote *SEM*. *$p < 0.05$; **$p < 0.01$; ***$p < 0.001$ (in green significance for comparison within ChR2 group and in black for comparison between ChR2 and YFP)

To further evaluate motivation to work for the laser-associated reward, animals were subjected to the progressive ratio task, in which effort increases throughout the session. ChR2 animals presented increased cumulative presses in the stim+ lever versus stim− lever session (Figure S3c; RM two-way ANOVA, session: $F_{(1.572, 59.73)} = 210.1$, $p < 0.0001$; group: $F_{(3, 38)} = 4.881$, $p = 0.0057$; Bonferroni's post hoc $t$ test, $t_{(17.78)} = 4.703$, $p = 0.0011$). This resulted in a significant effect of interaction between groups and session in the breakpoint (Figure 2d; $F_{(1, 19)} = 48.63$, $p < 0.001$). ChR2 animals presented a higher breakpoint in the stim+ lever than the stim− lever (Bonferroni's post hoc $t$ test, $t_{(19)} = 9.045$, $p < 0.001$), that is also significantly higher when compared to stim+ in YFP animals (Bonferroni's post hoc $t$ test, $t_{(38)} = 6.620$, $p < 0.0001$). ChR2 animals earn more rewards on the session for stim+ lever than the session for the stim− lever or YFP animals (Figure 2e; RM two-way ANOVA, $F_{(1, 19)} = 14.80$, $p = 0.0011$; Bonferroni's post hoc $t$ test, session: $t_{(19)} = 6.090$, $p < 0.0001$; group: $t_{(38)} = 3.020$, $p = 0.0090$).

Thus, activation of LDT–VTA projections is sufficient to increase preference for an otherwise equal reward and enhance motivation to work for that reward.

To further evaluate whether stimulation itself was an independent reinforcer, animals went through an extinction period, where pressing either lever did not yield any reward, but stim+ lever still originated laser stimulation. After a reminder session of FR1 schedule, both ChR2 and YFP animals decreased lever pressing in reward extinction conditions (Figure 2f; RM two-way ANOVA, session: $F_{(1.119, 42.51)} = 171.1$, $p < 0.0001$). This suggests that if previously paired with a reward, laser excitation of LDT–VTA inputs alone is ineffective in inducing preference.

## 3.3 | Optogenetic activation of LDT–VTA terminals is sufficient for place preference

To further understand the reinforcing properties of LDT–VTA terminal stimulation, we tested whether optogenetic modulation of LDT–VTA inputs induced place preference. We performed CPP (non-contingent) and RTPP (contingent) tests, pairing one chamber of each apparatus to laser stimulation (Figure 2g–k). Activating LDT–VTA terminals elicited place preference as shown by the increase in difference of time spent on the ON versus the OFF side of the CPP chamber of ChR2 animals (Figure 2h; $t$ test, $t_{(17)} = 5.604$, $p < 0.0388$).

In the RTPP, similarly, two-way ANOVA analysis showed a significant difference in the interaction between groups and chamber side (Figure 2k; two-way ANOVA, $F_{(1, 22)} = 30.09$, $p < 0.001$).

ChR2 animals spent more time in the laser-paired chamber when compared to the OFF side (Bonferroni's post hoc $t$ test, $t_{(22)} = 8.793$, $p < 0.001$); and compared to YFP animals ($t_{(44)} = 5.486$, $p < 0.001$). In both tests, YFP animals presented no preference for any chamber. Collectively, these results suggest that activation of LDT–VTA neurons triggers positive reinforcement.

## 3.4 | Optogenetic inhibition of LDT–VTA inputs decreases the value of a reward and motivational drive

We next performed LDT–VTA inhibition experiments using a similar strategy as before. We injected a cre-dependent halorhodopsin in the LDT (AAV5-Ef1a-DIO-NpHR-eYFP-WPRE-pA) and WGA-cre vector in the VTA (NpHR group; Figure 3a). In the two-choice task (Figure 3b), optogenetic inhibition of LDT–VTA terminals decreased preference for the stim+ lever in NpHR animals in comparison to stim− (Figure 3c; RM two-way ANOVA, session: $F_{(1.664, 63.24)} = 87.29$, $p < 0.0001$; group: $F_{(3, 38)} = 29.17$, $p < 0.0001$; Bonferroni's post hoc $t$ test, $t_{(10·10)} = 5.409$, $p = 0.0017$). This increase in preference for the stim− was attributed to an increase in the motivation for that lever, considering that the total lever presses on this task was not different between groups (Figure S3b). Motivational drive for either lever was again tested with the progressive ratio test (Figure 3d,e). Optical inhibition of LDT–VTA projections decreased motivation, since NpHR animals showed less cumulative presses for the stim+ lever (Figure S3d; RM two-way ANOVA, session: $F_{(1.330, 50.56)} = 265.6$, $p < 0.0001$; group: $F_{(3, 38)} = 6.409$, $p = 0.0013$). NpHR animals present a robust decrease in the breakpoint (Figure 3d; RM two-way ANOVA; $F_{(1, 19)} = 26.94$, $p < 0.0001$; Bonferroni's post hoc $t$ test, $t_{(19)} = 6.856$, $p < 0.0001$), including when compared to YFP animals (Bonferroni's post hoc $t$ test, $t_{(38)} = 2.941$, $p = 0.0111$). In agreement, the number of rewards earned during the progressive ratio task of NpHR animals is significantly less on the session for stim+ lever than the session for the stim− lever and between NpHR and YFP animals (Figure 3e; RM two-way ANOVA, $F_{(1, 19)} = 15.39$, $p = 0.0009$; Bonferroni's post hoc $t$ test, session: $t_{(19)} = 5.865$, $p < 0.0001$; group: $t_{(38)} = 3.082$, $p = 0.0076$).

In reward extinction conditions, both groups decreased instrumental responding for both levers since the first trial of extinction conditions (Figure 3f; RM two-way ANOVA, session: $F_{(1.073, 40.76)} = 112.4$, $p < 0.0001$).

In the CPP test (Figure 3g), optical inhibition of LDT–VTA inputs did not induce statistically significant place preference/avoidance for any of the chambers (Figure 3h; $t$ test, $t_{(22)} = 1.859$, $p = 0.076$). LDT–VTA optical inhibition caused decreased preference for the ON chamber in the RTPP (Figure 3i–k; two-way ANOVA, interaction: $F_{(1, 22)} = 31.74$, $p < 0.0001$). NpHR animals spent less time in the laser-paired side of the chamber than the OFF side (Bonferroni's post hoc $t$ test, $t_{(22)} = 8.324$, $p < 0.0001$); and also in comparison to YFP animals (Bonferroni's post hoc $t$ test, $t_{(44)} = 5.634$, $p < 0.0001$).

## 3.5 | Differential recruitment of VTA and NAc neuronal populations

We determined the activation pattern of different neuronal populations after the PR task in YFPs, ChR2, and NpHR groups, while optically manipulating LDT terminals in the VTA during the reward period of the PR test, in animals working for either stim− lever or

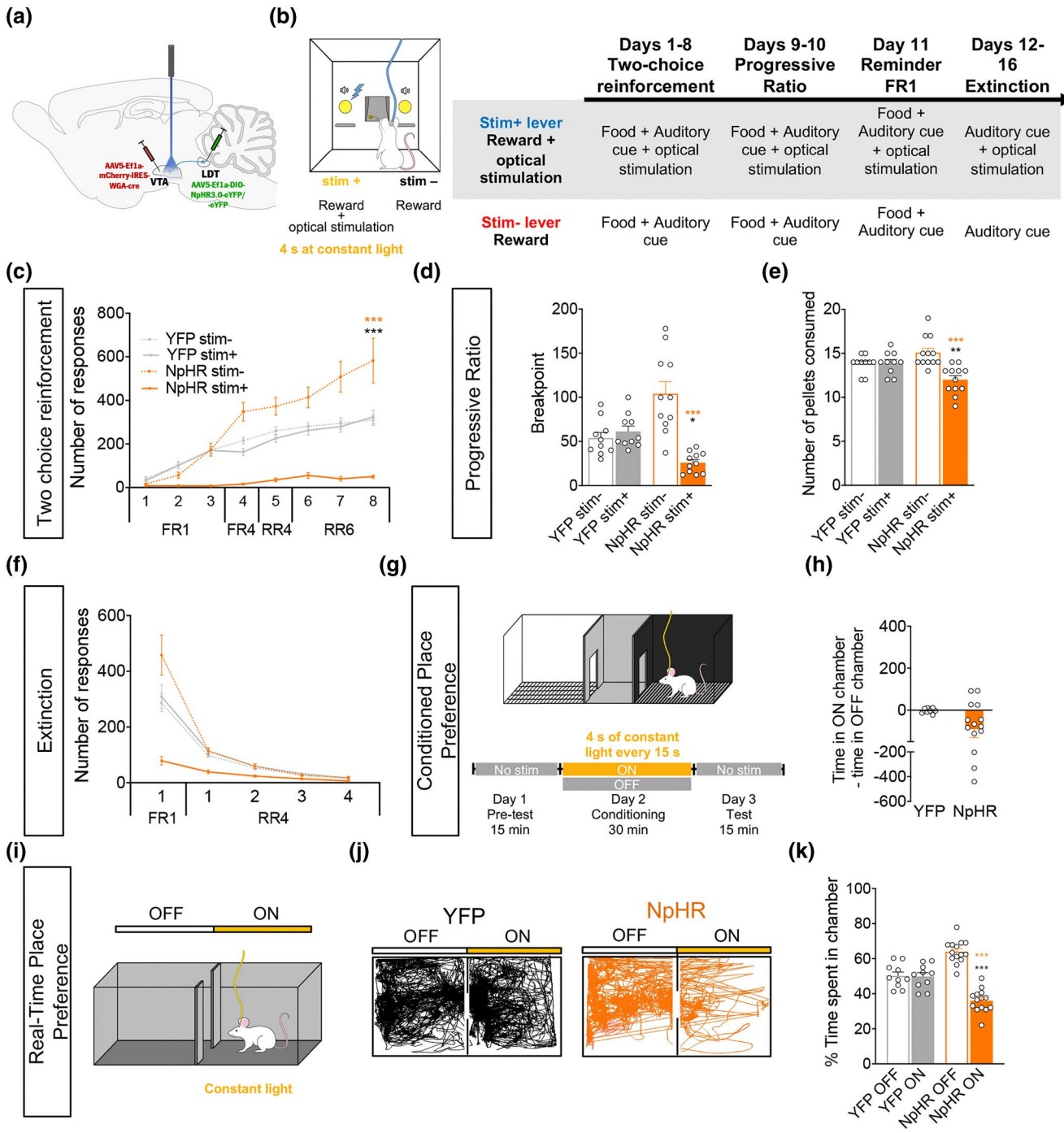

**FIGURE 3** Optogenetic inhibition of LDT terminals in the VTA decreases motivation. (a) Strategy used for LDT–VTA projection optogenetic inhibition during behavior. (b) Schematic representation of the two-choice task. Pressing stim− lever yields one food pellet and pressing stim+ lever delivers one pellet + optical inhibition of LDT–VTA inputs (4 s of constant yellow laser at 10 mW). (c) Time-course representation of the responses in NpHR ($n = 11$) and YFP ($n = 10$) rats. Optogenetic inhibition of LDT–VTA terminals shifts preference for the non-stimulated lever (stim−) in NpHR animals, but no preference is observed in YFP group. (d) Decrease in breakpoint for stim+ lever in NpHR animals. (e) Total number of rewards earned during progressive ratio of NpHR and YFP rats. NpHR worked less to earn rewards for the stim+ lever. (f) In pellet extinction conditions, both groups decrease responses for both levers. (g) CPP and (i) RTPP paradigms, in which one chamber is associated with laser stimulation (ON side). (h) Difference of total time spent in the OFF and ON sides in YFP ($n = 10$) and NpHR ($n = 14$) groups. (j) Representative tracks for a NpHR and a YFP animal during the RTPP. (k) Percentage of time spent on the ON and OFF sides, showing preference for the side associated with no stimulation. Error bars denote *SEM*. *$p < 0.05$; **$p < 0.01$; ***$p < 0.001$ (in orange significance for comparison within NpHR group and in black for comparison between NpHR and YFP)

stim+ lever (Figure S4). We next evaluated the density of positive c-fos$^+$ cells that were either ChAT$^+$ (cholinergic neurons) or ChAT$^-$ in the LDT (Figure 4a,b), TH$^+$ or TH$^-$ in the VTA (Figure 4e,f), and D1R$^+$, D2R$^+$ or ChAT$^+$ in the NAc (Figure 4j,k).

In the LDT, ChR2 stim+ group displayed an increase in c-fos$^+$ cells in comparison to ChR2 stim− or YFP groups (Figure S4a; one-way ANOVA, $F_{(3, 16)} = 10.98$, $p = 0.0004$; ChR2 stim+ vs. stim−: post hoc $t$ test, $t_{(16)} = 5.506$, $p = 0.0003$; ChR2 stim+ vs. YFP stim+: post hoc $t$ test, $t_{(16)} = 4.002$, $p = 0.0062$). In agreement, ChR2 stim+ group presented an increase in c-fos$^+$/ChAT$^+$ (Figure 4c; two-way ANOVA, $F_{(3, 16)} = 6.697$, $p = 0.0039$) in comparison to ChR2 stim− (ChAT: post hoc $t$ test, $t_{(32)} = 6.563$, $p < 0.0001$), or in comparison to YFP animals stimulated animals (ChAT: post hoc $t$ test, $t_{(32)} = 6.535$, $p < 0.0001$). No effect was observed in c-fos$^+$/ChAT$^-$ cells.

No significant differences were observed in c-fos$^+$ cell density of NpHR or YFP-inhibition groups, independently of the lever (Figure S4b). Nevertheless, NpHR stim+ group showed a specific decrease in cell density of c-fos$^+$ChAT$^+$ in LDT, when compared to the NpHR stim− group (Figure 4d; two-way ANOVA, $F_{(3, 16)} = 14.28$, $p < 0.0001$; post hoc $t$ test, $t_{(32)} = 4.305$, $p = 0.0009$). Recruitment of ChAT$^-$ cells was increased in NpHR stim+ animals when compared to NpHR stim− (Bonferroni's post hoc $t$ test, $t_{(32)} = 5.402$, $p < 0.0001$) or YFP stim+ animals (Bonferroni's post hoc $t$ test, $t_{(32)} = 6.207$, $p < 0.0001$).

Next, we analyzed the recruitment of VTA cells in medial and lateral subregions (Figure S4c; two-way ANOVA, $F_{(1.923, 7.692)} = 11.28$, $p = 0.0054$). ChR2 stim+ group did not present significant differences in the number of c-fos$^+$ cells in medial VTA in comparison to ChR2 stim− or YFP groups. Conversely, ChR2 stim+ group displayed an increase in c-fos$^+$ cells in the lateral VTA in comparison to ChR2 stim− (Bonferroni's post hoc $t$ test, $t_{(4)} = 10.57$, $p = 0.0027$) and YFP group (Bonferroni's post hoc $t$ test, $t_{(4)} = 6.545$, $p = 0.0169$) (Figure S4c).

We observed a significant increase in c-fos$^+$/TH$^+$ cells in the ChR2 stim+ group in the lateral VTA in comparison to ChR2 stim− (Bonferroni's post hoc $t$ test, $t_{(32)} = 1.422$, $p < 0.0001$) and YFP group (Bonferroni's post hoc $t$ test, $t_{(32)} = 1.799$, $p < 0.0001$) with no significant differences on c-fos$^+$/TH$^-$ cells; this effect was not observed in the medial VTA.

NpHR stim+ group did not present any significant differences in the number of c-fos$^+$ TH$^+$ cells, or c-fos$^+$TH$^-$ cells in medial or lateral VTA (Figure 4h).

Interestingly, there was a positive correlation between individual performance in the progressive ratio task and the recruitment of TH$^+$ cells in the medial or lateral VTA (Figure S5a,b; Medial VTA: Pearson's $r = 0.5070$, $p = 0.0008423$; Lateral VTA: Pearson's $r = 0.71170$, $p = 2.621e^{-007}$).

Considering that LDT neurons preferentially target NAc-projecting cells in the VTA (Forster & Blaha, 2000; Lammel et al., 2012; Omelchenko & Sesack, 2005), we analyzed the number of recruited cells in NAc subregions, core and shell after PR test for either lever (Figure S4e,f). Chr2 stim+ animals showed a significant increase in c-fos$^+$ cells in the core and shell (Bonferroni's post hoc $t$ test, core: $t_{(4)} = 5.471$, $p = 0.0373$, shell: $t_{(4)} = 4.858$, $p = 0.0498$).

LDT–VTA NpHR animals did not present significant changes in recruited cells.

We next asked if NAc cells were differentially activated between experimental groups. Thus, we quantified the number of double positive cells for c-fos and dopamine receptor D1, dopamine receptor D2, or ChAT in the NAc core and shell subregions (Figure 4i–l).

c-fos ChR2 stim+ animals showed a significant increase in the recruitment of D1$^+$ cells when compared to ChR2 stim− or YFP animals, in either core or shell subregions of the NAc (Figure 4m; one-way ANOVA, core: $F_{(3, 16)} = 31.58$, $p < 0.0001$; shell: $F_{(3, 16)} = 16.99$, $p < 0.0001$). No alterations were found in the activation of D2R$^+$ or ChAT$^+$ cells.

NpHR stim+ animals presented an increase in c-fos$^+$/D2R$^+$ in the NAc core and shell subregions, in comparison to other groups (Figure 4n; one-way ANOVA, core: $F_{(3, 16)} = 16.83$, $p < 0.0001$; shell: $F_{(3, 16)} = 4.952$, $p = 0.0128$). No major differences were found in the number of activated D1$^+$ or ChAT$^+$ cells.

There was a positive correlation between individual performance in the progressive ratio task and the recruitment of D1$^+$ cells in both the NAc core and shell subregions (Figure S5c,f; NAc core: Pearson's $r = 0.5702$, $p = 0.0001224$; NAc shell: Pearson's $r = 0.4253$, $p = 0.006222$). No significant correlation was found for other neuronal populations.

Interestingly, recruitment of cells in the NAc core was positively correlated with VTA c-fos in the medial part (Figure S5i; Pearson's $r = 0.2147$, $p = 0.0026$), and the number of c-fos$^+$ cells in the NAc shell was positively correlated with c-fos$^+$ cells in the lateral VTA (Figure S5j; Pearson's $r = 0.2547$, $p = 0.0009$).

## 4 | DISCUSSION

In this work, we show that optical activation of LDT–VTA terminals shifts and amplifies preference for a laser-paired reward in comparison to an otherwise equal reward, and that optical inhibition led to an opposite effect. If the reward was omitted, animals abolished preference, suggesting that LDT–VTA stimulation adds/decreases value to the stimulation-paired reward. Moreover, LDT–VTA optical activation increases motivational drive, and the reverse occurs when we inhibited these inputs. In addition to these novel data, we further confirmed previous evidence showing that LDT–VTA optical activation induces robust preference in the conditioned and RTPP tests (this study and Lammel et al., 2012; Steidl, Wang, et al., 2017; Xiao et al., 2016); while optical inhibition induces aversion (this study). Interestingly, LDT–VTA activation increased recruitment of lateral VTA dopamine neurons and D1 neurons from the NAc; whereas inhibition preferentially recruited D2 neurons.

In this work, to evaluate if LDT–VTA neurons could increase the value of a given reward, we performed the two-choice task, in where rats are allowed to press a lever to obtain a food pellet or a similar food pellet paired with laser stimulation. ChR2 animals progressively press more stim+ lever, that is, LDT–VTA activation enhanced preference of an otherwise equal reward. The PR test was used to

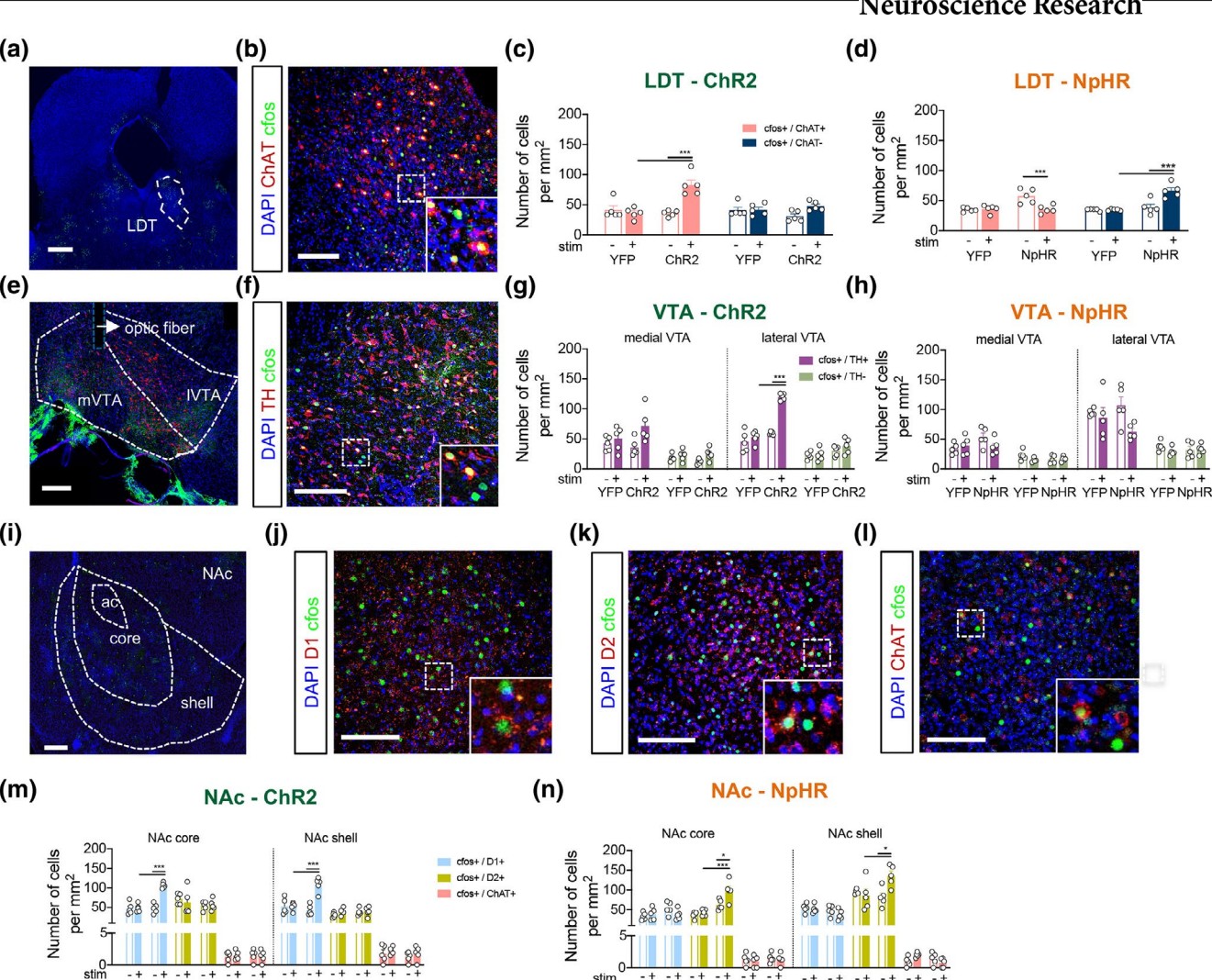

**FIGURE 4** Recruitment of neurons in the LDT, VTA, and NAc after optical modulation of LDT–VTA terminals. (a) Representation of LDT visualized at bregma = −8.5 AP, scale bar = 250 µm. (b) Immunofluorescence of the LDT with staining for cell nuclei (DAPI, blue), ChAT (red), and c-fos (green), scale bar = 100 µm, inset of positive cell. (c) Density of c-fos and ChAT double positive cells in ChR2 and YFP rats or (d) NpHR and YFP after PR performance on the stim+ or stim− lever. (e) Representation of medial and lateral VTA subregions visualized at bregma = −5.2 AP, scale bar = 100 µm. (f) Immunofluorescence of the VTA, staining for cell nuclei (DAPI, blue), tyrosine hydroxylase (TH, red), and c-fos (green), scale bar = 100 µm, inset of positive cell. (g) Density of c-fos and TH double positive cells in ChR2 and YFP rats or (h) NpHR and YFP after PR performance on the stim+ or stim− lever. There is an increase in the number of c-fos$^+$/TH$^+$ cells in the lateral VTA after LDT–VTA optical activation, whereas optical inhibition shows no significant differences in the number of recruited cells when compared to YFP animals. (i) Representation of NAc core and shell subregions. (j) Immunofluorescence of the NAc, staining for cell nuclei (DAPI, blue), c-fos (green) and D1R (red), (k) DR2 (red) or (l) ChAT (red), scale bar = 100 µm. Inset of positive cells in each staining. (m) Density of c-fos and D1R, DR2 or ChAT double positive cells in ChR2 and YFP rats or (n) NpHR and YFP rats after PR performance on the stim+ or stim− lever. There is an increase in the number of c-fos$^+$/D1R$^+$ cells in both NAc core and shell subregions after LDT–VTA optical activation, whereas optical inhibition appears to recruit mostly D2R cells. No significant differences were found in the number of c-fos$^+$/ChAT$^+$ cells. For all cell countings: nstim+ = 5; nstim− = 5 in each group. Error bars denote *SEM*. **$p < 0.01$; ***$p < 0.001$

evaluate the motivation of animals to work for a particular reward. Importantly, animals presented a substantial increase in their motivation to work for the LDT–VTA stim+ lever, suggesting that excitation of these inputs amplified the motivational attractiveness of its paired reward representation, raising its value. In contrast, when reward was paired with LDT–VTA inhibition, preference and motivational drive was shifted toward the stim− lever.

Interestingly, removing the reward (pellet) dramatically decreased lever pressing, hinting that LDT–VTA excitation added value to that reward (whereas inhibition decreased it). However, these results are challenging to conciliate with the fact that LDT–VTA terminal activation also appears to be rewarding per se, since it induced place preference in both the CPP and real-time preference tests (as observed here and in Lammel et al., 2014; Steidl & Veverka, 2015;

Steidl, Wang, et al., 2017; Xiao et al., 2016). Moreover, optogenetic inhibition of LDT–VTA inputs induced avoidance behavior in a real-time preference test (this study). These findings highlight the complexity of reward behaviors, and the importance of the context and associative learning in the process. Considering the two-choice task, animals are food deprived and their main goal is to press the lever to obtain a food pellet, so this goal-directed behavior involves a clear action-outcome. Once the pellet (outcome) is removed, the instrumental action is no longer reinforced, so animals quit responding. On the other hand, the place preference tests are used to evaluate context associations based on the rewarding (or aversive) properties of an unconditioned stimulus (drugs or optical stimulation). It is important to refer that in the preference tests, animals do not receive any outcome in the test day, since the preference is based on a previous emotionally relevant association. These findings resemble a previous study in which optogenetic stimulation of LDT-NAc terminals specifically added value to a laser-paired external reward, but when that reward was absent, preference was no longer observed (Coimbra et al., 2019). Other studies have also originated puzzling effects in different tests that evaluate the rewarding properties of a stimulus. For example, optogenetic stimulation of GABAergic projections from the CeA to ventromedial PFC increases motivation to get a food reward and induces place preference in the RTPP, but is not able to induce self-stimulation behavior (Seo et al., 2016), in line with the notion that a rewarding stimulus can act as reinforcer in one context (RTPP) but not in others (self-stimulation).

Our data suggest that the LDT can convey positive reinforcing signals to the VTA, but also to the NAc (Coimbra et al., 2019), altering reward value. Activation of LDT–VTA axons produced an overall focused effect and increase in motivation that can be the result of inducing dopamine release from mesolimbic neurons into the NAc (Forster & Blaha, 2000; Steidl, O'Sullivan, et al., 2017). Indeed, our electrophysiological recordings showed that LDT terminal activation increased firing rate of VTA putative dopamine cells while decreasing the activity of putative GABAergic cells. These electrophysiological findings are in agreement with studies showing that increased VTA dopaminergic activity triggers positive reinforcement (Beier et al., 2015; Beierholm et al., 2013; Coimbra et al., 2017; Hamid et al., 2016; Roitman et al., 2004; Steidl & Veverka, 2015; Tsai et al., 2009), which we also observed in our study.

In this work, we also evaluated neuronal recruitment during a behavioral proxy of motivation—the PR test—in the LDT, VTA, and NAc. ChR2 animals showed an increase in the recruitment of ChAT[+] cells in the LDT, accompanied by an increase in c-fos[+]TH[+] cells in the lateral VTA of ChR2 animals, suggesting an important role for LDT excitatory cholinergic inputs to VTA in motivation. In agreement, there was a significant positive correlation between TH[+] recruited cells and motivational drive (breakpoint) in both the lateral and medial VTA. In fact, previous reports showed that cholinergic projections from the LDT are preferentially directed to the lateral region of the VTA, that in turn projects to the lateral shell subregion of the NAc (Lammel et al., 2012), a subregion important for motivated behaviors (Pascoli et al., 2015; Reynolds & Berridge, 2002; Tsai et al., 2009).

Interestingly, NpHR animals also presented an increase in c-fos[+] cells in the LDT, which mirrors the observed behavioral effects. This recruitment change is probably not due to the direct LDT terminal optical inhibition, but rather a combined effect of multiple signals to the LDT underlying the observed differences in motivation.

For long it is known that VTA-NAc dopamine input activation is sufficient to produce reward in rats (Holmes & Fam, 2013; Ikemoto & Panksepp, 1999; Steidl, O'Sullivan, et al., 2017; Steinberg et al., 2014). In fact, distinct dopaminergic sub-populations with anatomically and functionally distinct populations, project to different NAc subdivisions (shell and core) (Beier et al., 2015; Lammel et al., 2008, 2011; Yang et al., 2018) that may exert opposing influences on motivated behaviors (Aragona et al., 2008; Bassareo et al., 2002; Dreyer et al., 2016). Here, we were able to observe a positive correlation of recruited cells in the NAc core and NAc shell with increased c-fos[+] cells in the medial and lateral VTA, respectively, highlighting the role of modulatory inputs from the LDT to VTA in reward-related behaviors.

We found that in ChR2 animals, there was an increase in c-fos[+]/D1[+] cells, and that there was a positive correlation between c-fos[+]/D1[+] and breakpoint, which is in line with the pro-motivational role attributed to this neuronal population (Francis et al., 2015; Hikida et al., 2013; Ikemoto et al., 1997; Kravitz et al., 2012; Lobo et al., 2010; Soares-Cunha et al., 2016, 2020).

NpHR animals displayed a decrease in c-fos[+]/ChAT[+] cells in the LDT, with no significant changes in the VTA. Surprisingly, these animals presented an increase in c-fos[+]D2[+] cells in the NAc core and shell. D2[+] cells have been classically associated with negative reinforcement and aversion (Al-Hasani et al., 2015; Kravitz et al., 2012; Lobo et al., 2010; Volman et al., 2013), although this effect appears to depend on the pattern of activity of this neuronal population (Kupchik et al., 2015; Soares-Cunha et al., 2016, 2020). In agreement, we observed that LDT–VTA optical inhibition decreases preference for the laser-associated chamber in the RTPP.

One important consideration is that the LDT also sends direct projections to the NAc (Coimbra et al., 2019; Dautan et al., 2014), although to date there is no evidence about preferential innervation of LDT to NAc D1 or D2 neurons. Thus, the observed changes in recruitment neurons in the NAc may be a direct effect of LDT modulation rather than via VTA. We believe it is now crucial to develop further studies to unveil how signals in the LDT–VTA–NAc triangle are orchestrated to drive behavior. Who is the *master* and in which stage of behavior?

In sum, this work shows that activation of LDT–VTA specific inputs is sufficient to induce positive reinforcement and increase preference for an otherwise equal reward, whereas suppression of activity has an opposite effect. Moreover, LDT–VTA activation/inhibition boosts/decreases motivation. These behavioral findings are supported by electrophysiological and c-fos immunofluorescence correlates in downstream target regions, namely the VTA and NAc. Since the LDT sends direct projections to both the VTA and the NAc (Beier et al., 2015; Coimbra et al., 2019; Dautan et al., 2014; Dautan, Souza, et al., 2016; Lammel et al., 2012; Omelchenko &

Sesack, 2005), additional studies are needed to understand how LDT signals are integrated in the mesolimbic circuitry to drive positive/negative reinforcement.

## DECLARATION OF TRANSPARENCY

The authors, reviewers and editors affirm that in accordance to the policies set by the *Journal of Neuroscience Research*, this manuscript presents an accurate and transparent account of the study being reported and that all critical details describing the methods and results are present.

### ACKNOWLEDGMENTS
The authors thank the Animal facility technicians for the technical support with rodent husbandry.

### CONFLICT OF INTEREST
The authors declare no conflict of interest.

### AUTHOR CONTRIBUTIONS
All authors read and approved the manuscript. *Conceptualization*, B.C. and A.J.R.; *Methodology*, B.C. and A.J.R.; *Formal Analysis*, B.C.; *Investigation*, B.C., A.V.D., C.S-C., and R.C.; *Resources*, A.J.R. and N.S.; *Writing – Original Draft*, B.C. and A.J.R.; *Writing – Review & Editing*, B.C., L.P., N.S., and A.J.R.; *Visualization*, B.C. and A.J.R.; *Supervision*, A.J.R.; *Funding Acquisition*, A.J.R. and N.S.

### PEER REVIEW
The peer review history for this article is available at https://publons.com/publon/10.1002/jnr.24931.

### DATA AVAILABILITY STATEMENT
The data that support the findings of this study are available from the corresponding author upon request.

### ORCID
*Bárbara Coimbra* [iD] https://orcid.org/0000-0003-1737-2268
*Ana Verónica Domingues* [iD] https://orcid.org/0000-0003-2605-8254
*Carina Soares-Cunha* [iD] https://orcid.org/0000-0001-9470-644X
*Raquel Correia* [iD] https://orcid.org/0000-0002-5708-858X
*Luísa Pinto* [iD] https://orcid.org/0000-0002-7724-0446
*Nuno Sousa* [iD] https://orcid.org/0000-0002-8755-5126
*Ana João Rodrigues* [iD] https://orcid.org/0000-0003-1968-7968

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

## SUPPORTING INFORMATION

Additional Supporting Information may be found online in the Supporting Information section.

Transparent Peer Review Report

Transparent Science Questionnaire for Authors

**FIGURE S1** Confirmation of optic fiber implantation for behavioral experiments. (a) Optic fiber placement in the VTA for YFP (activation—Stimulation with blue laser), ChR2, YFP (inhibition—Stimulation with yellow laser) and NpHR animals

**FIGURE S2** Experimental design for electrophysiological and behavioral studies. (a) For set 1 of experiments, for electrophysiological recordings, 30 days after surgery for virus injections in LDT and VTA, animals were subjected to anesthesia in order to place a recording electrode together with an optic fiber in VTA, to stimulate LDT terminals. Confirmation of electrode placement was confirmed after the experiment. (b) For behavioral experiments, 30 days after surgery for virus injections in LDT and VTA, and cannula implantation in the VTA (n of animals indicated in figure), animals from all four groups of set 2 performed the real-time place preference (1 day). In the following week, the same cohort performed the conditioned place preference (3 days), Operant behavior in the two-choice schedule of reinforcement started on the 20th day where animals are able to choose between pressing a lever for a reward and optical stimulation (stim+) or another lever for a reward alone (stim−), with increasing effort, until a random ratio 6 (RR6). This is followed by two sessions of a progressive ratio (PR) task for each lever. Then, animals performed the extinction phase of the two-choice reinforcement behavior, where the reward was omitted for both levers but optical

stimulation remained for the stim+ lever, for 4 days. For operant behavior the number of animals is reduced, since we had to exclude animals that lost the fiber implant. (c) Experimental design for c-fos induction. After the two-choice reinforcement operant protocol, animals performed a reminder session and, on the next day, followed with a PR session for c-fos activation studies. In order to distinguish cell recruitment for task performance for a pellet alone or a pellet associated with optical stimulation, groups were divided to perform PR for stim+ or for the stim− lever (*n* of animals indicated in figure)

**FIGURE S3** Effect of optogenetic modulation of LDT–VTA terminals on lever pressing performance in the two-choice paradigm and progressive ratio. Total number of lever presses during all sessions of the two-choice paradigm for (a) ChR2 and YFP rats or (b) NpHR and YFP rats on the stim+ or stim− lever. (c) Cumulative presses performed during the progressive ratio task show that ChR2 animals press more on stim+ lever. (d) Cumulative presses performed during the progressive ratio task show that NpHR animals press more on stim− lever. Error bars denote *SEM*. \*\**p* < 0.01; \*\*\**p* < 0.001 (green significance for comparison for ChR2 stim+ and stim− groups and in black for comparison between ChR2 and YFP; orange significance for comparison within NpHR stim+ and stim− groups and in black for comparison between NpHR and YFP)

**FIGURE S4** Recruitment of total c-fos$^+$ cells in the LDT, VTA and NAc regions after Progressive Ratio performance. Quantification of c-fos in the LDT of (a) ChR2 and YFP rats or (b) NpHR and YFP rats after PR performance on the stim+ or stim− lever. Quantification of c-fos in the VTA of (c) ChR2 and YFP rats or (d) NpHR and YFP rats after PR performance on the stim+ or stim− lever. Quantification of c-fos

in the NAc of (e) ChR2 and YFP rats or (f) NpHR and YFP rats after PR performance on the stim+ or stim− lever. nstim+ = 5; nstim− = 5 in each group. Error bars denote *SEM*. \**p* < 0.05; \*\**p* < 0.01; \*\*\**p* < 0.001 (green significance for comparison for ChR2 stim+ and stim− groups and in black for comparison between ChR2 and YFP; orange significance for comparison within NpHR stim+ and stim− groups and in black for comparison between NpHR and YFP)

**FIGURE S5** Recruitment of VTA dopaminergic cells is positively correlated with motivational drive. Pearson's correlation between individual breakpoint of ChR2, NpHR or YFP animals and the number of c-fos$^+$/TH$^+$ cells in the (a) medial and (b) lateral VTA. There is a positive correlation between the number of TH recruited cells in the VTA and individual motivational drive (given by the breakpoint in the PR task). (c–h) Pearson's correlation between individual breakpoint of ChR2, NpHR or YFP animals and the number of c-fos$^+$/D1$^+$ cells, c-fos$^+$/D2$^+$ cells or c-fos$^+$/ChAT$^+$ cells in the NAc core (c–e) and shell (f–h). (i,j) Positive Pearson's correlation between c-fos$^+$ cells in the medial and lateral VTA and c-fos$^+$ in the NAc core and shell subregions, respectively. Error bars denote *SEM*. \*\**p* < 0.01; \*\*\**p* < 0.001

