## [Transparent Peer Review Report · Journal of Neuroscience Research]

Laterodorsal tegmentum-Ventral Tegmental Area projections encode positive reinforcement signals
Coimbra, Bárbara; Domingues, Ana Verónica; Soares-Cunha, Carina; Correia, Raquel; Pinto, Luísa;
Sousa, Nuno; Rodrigues, Ana João

Review timeline: Submission date: 8 March 2021
Editorial Decision: Minor Modification (15 April 2021)
Revision Received: 31 May 2021
Accepted: 12 July 2021

Editor 1: Bertrand Lambolez
Editor 2: Cristina Ghiani
Reviewer 1: Jerome Baufreton
Reviewer 2: Jacques Barik
Reviewer 3: Jennifer Kaufling

1st Editorial Decision

Decision letter

Dear Dr Coimbra:

Thank you for submitting your manuscript to the Journal of Neuroscience Research. We've now received the reviewer feedback and have appended those reviews below. I'm glad to say that the reviewers are overall very enthusiastic and supportive of the study. They did raise some concerns and made some suggestions for clarification, but I expect that these points should be relatively straightforward to address. If there are any questions or points that are problematic, please feel free to contact me. I am glad to discuss.

We ask that you return your manuscript within 30 days. Please explain in your cover letter how you have changed the present version and submit a point-by-point response to the editors' and reviewers' comments. If you require longer than 30 days to make the revisions, please contact Dr Cristina Ghiani (cghiani@mednet.ucla.edu). To submit your revised manuscript: Log in by clicking on the link below <https://wiley.atyponrex.com/submissionBoard/1/14c1d00e-3baa-45da-bbf0-b41c5ce5511d/current>

(If the above link space is blank, it is because you submitted your original manuscript through our old submission site. Therefore, to return your revision, please go to our new submission site here (submission.wiley.com/jnr) and submit your revision as a new manuscript; answer yes to the question "Are you returning a revision for a manuscript originally submitted to our former submission site (ScholarOne Manuscripts)? If you indicate yes, please enter your original manuscript's Manuscript ID number in the space below" and including your original submission's Manuscript ID number (jnr-2021-Mar-9632) where indicated. This will help us to link your revision to your original submission.)

The journal has adopted the "Expects Data" data sharing policy, which states that all original articles and reviews must include a Data Availability Statement (DAS). Please see <https://authorservices.wiley.com/author-resources/Journal-Authors/open-access/data-sharing-citation/data-sharing-policy.html#standardtemplates> for examples of an appropriate DAS. Please include the DAS in the manuscript as well.

Thank you again for your submission to the Journal of Neuroscience Research; we look forward to reading your revised manuscript.

Best Wishes,

Dr Bertrand
Associate Editor, Journal of Neuroscience Lambolez
Research

Dr Cristina
Editor-in-Chief, Journal of Neuroscience Ghiani
Research

Editors Comments to the Author:

Please mention in the title and/or abstract that the work was done only in male rats, also discuss this as a limitation.

SEX DIFFERENCES

The National Institutes of Health now mandates the inclusion of sex as a biological variable. To conform with this mandate, the Journal of Neuroscience Research has established new policy (please see our editorial: <http://onlinelibrary.wiley.com/doi/10.1002/jnr.23979/full>) requiring all authors to ensure proper consideration of sex as a biological variable. Please ensure that: 1) Any paper utilizing subjects of one sex state the sex of the sample in the title and abstract; 2) The number of samples/subjects of each sex used in the research must be clearly stated in the methods section; 3) The inability for any reason to study sex differences where they may exist should be discussed as a study limitation. We are also encouraging authors to report exploratory analyses of potential sex differences in studies not explicitly designed to address them.

Please submit the figures as separate files.

Reviewer: 1

Comments to the Author

In this study Coimbra and al., investigate the involvement of LDT-VTA circuit in positive reinforcement signals encoding. Using viral vector-based approaches in rat, they demonstrate that the stimulation (ChR2 experiments) of the LDT-VTA pathway increases the motivation whereas whereas inhibition (NpHR experiments) has the opposite effect. They also show that stimulation of LDT-VTA pathway is sufficient for place preference while optical inhibition induces aversion.

They performed appropriate control experiments including electrophysiological validation of their strategy (eg selective manipulation of dopaminergic neurons of the VTA) and behavioral control groups expression only the fluorescent reporter YFP.

The last part of the manuscript investigates the cell types involved in the behavioral effects and provide convincing evidences of the recruitment of lateral VTA dopamine neurons and D1-neurons from nucleus accumbens core and shell in the ChR2 experiments, whereas in LDT-VTA NpHR animals D2-neurons appear to be preferentially recruited.

This is elegant and well designed study and the conclusions drawn by the authors are fully supported by the data. The manuscript is clearly written and the figures extremely well constructed. The results are new and of great interest for the field. I have no criticism on this study.

Reviewer: 2

Comments to the Author

In this manuscript the authors assessed the role of the laterodorsal tegmentum (LDT) projections to the ventral tegmental area (VTA) and their role on reward processing. Optical manipulations of the LDT-VTA projections are shown to influence reward value and the behavior related to it. The demonstrations have been carried out on an operant task with food reinforcers, as well as place conditioning tasks, and were accompanied by immuno-based assays showing correlation between c-fos expression in the LDT, VTA and nucleus accumbens (NAcc) and the behavioral results.

Overall, the work represents an interesting contribution on the participation of LTD on the reward system, but a few observations should be addressed:

Minor question: Page 15, line 37: The names of the figures should be corrected (Fig. 2c; line 52: Fig. 2d).

Major questions:

[1] The assertion that the inhibition of the LDT-VTA projection (NpHR experiment) conveys a decrease in motivation.

As opposed to the previous experiment (with ChR2), in which animals worked more for the paired (food pellet+stimulation) reinforcer than the control group, a decreased number of stim+ paired lever presses may not necessarily convey decreased motivation in a two choice paradigm. That is: the decrease on stim+ presses (if, say, compensated by an increase in stim- presses) may rather reflect a bias in lever preference, not a change in motivational status.

It is important to show the total number of rewards obtained by each group. Despite the preference being so far convincing, a decrease in motivation would be more convincingly demonstrated by a decrease in the total number of rewards earned.

[2] Another relevant question concerns the seemingly conflicting results obtained from ChR2 experiments.

Specifically, the fact that stim+ lever was subjected to extinction, even though the same stimulation induced place preference. On this regard, the number of magazine entries will give a useful insight on potential differences in the attribution of value to the food itself, as magazine approach is a parameter sensitive to changes in the value of the outcome. Also, the discussion should mention the behavioral and neuroanatomical substrates required for place preference and instrumental conditioning, which may give insights regarding these results. [3] Finally, a word should be addressed to the results obtained by c-fos analyses in the LDT within the NpHR groups. That is, YFPstim+ and YFPstim- show similar patterns of c-fos expression; however, the NpHRstim-group showed a seemingly increase in c-fos (at least enough to present statistical significance when compared to the NpHRstim+ group). Given that no optical stimulation has been done, should one not expect c-fos expression in this group to be similar to the YFP groups? Authors should discuss the potential causes for this difference.

Reviewer:

3

Comments to the Author
In this study, the authors investigate the impact of LDT -VTA projections on natural reward related behaviours. To do so, they successfully combine viral optogenetic approaches (excitation and inhibition of LDT-VTA inputs) and classic reward related tasks including, progressive ratio task and place preference tests. They use electrophysiological recordings to assess the functional efficiency of the viral approaches and use cfos staining to correlate some behavioural results with cfos neuronal activation. While the manuscript is highly interesting, well conducted and well written and I have some suggestions that could improve the methodology and the interpretation of results.

- Regarding viral approaches, the authors should demonstrate the specificity of the viral infections. According to the viral strategy used, LDT neurons that express opsins (hChR2 or NpHR) should also express WGA. In that case, in the LDT, each green cell body (YFP) must also be red (mcherry). This is not shown or discussed in the results. Fig 1 b and c are not enough (only YFP in LDT). The authors should also discuss the efficacy, i.e. the number or proportion of LDT neurons that express opsins.

-Regarding electrophysiological approaches, some important information is missing. For the optotagging approach, the authors write, "Neurons showing a firing rate increase or decrease by more than 20% from the mean frequency of the baseline period were considered as responsive."

- What is the duration of the baseline period?
- Why use 20% increase or decrease in activity?
- What are the responses to each 10ms pulse? Do the neurons follow 80 stimulations at 20hz? The traces in fig 1 g and I are not sufficient. The authors should use more appropriate time resolution to respond to above questions.

- Do panel's g and I in figure 1 represent a single stimulation period (80 stims) in one animal, or several stimulation periods in several animals? In the second case, how the authors did the normalization. Error bars are also missing if these traces correspond to an average. What those traces represent is unclear.

- In the results section, the authors mentioned that "Activation of LDT terminals evoked excitatory responses in 79% (55 out of 70 cells) of putative VTA dopaminergic (pDAergic) neurons and inhibitory responses in 71% (10 cells out of 14) putative VTA GABAergic (pGABAergic) neurons recorded. These numbers are high. Are they supported by literature? The similar question should be answered for optical inhibition (large proportion of responding VTA neurons).

-p. 15 line 24 "To test the impact of LDT-VTA manipulation in behavior, we unilaterally activated these inputs in freely moving rats during a two-choice instrumental task (Fig. 2a)" fig 2a is a schematic of the viral /optogenetic stimulation procedure. The behavioural approach seems to be represented in fig 2b. There is a gap between the main text and the panels in figure 2. The letter does not match the graphic. The same is present in the legend of the figure.

-p. 16 lines 44-45 "Similarly, two-way ANOVA analysis showed a significant difference in the interaction between groups and chamber side (Fig. 2j; two-way ANOVA, $F(1, 20) = 26.93, p < 0.001$)." No corresponding graph in the figure 2

-p. 17 line 14. Nac instead of VTA?

- p. 17 and figure 3. Same as for figure 2: the drawing / graph represented in the figure do not correspond to

the description in the main text. The lettering does not match the graphs.

-p18line 1-10. Supplemental fig 2 corresponds to the experimental design of behavioural studies and not to the activation pattern of neuronal populations after the PR tasks as described in the main text. Is that the right figures? Is this the right figure? If so, the authors should adapt the description of the figure in the main text or the title of the legend. The description of the different panels of the figure is missing in the main text.

Author response

Prof. Cristina A. Ghiani and J. Paula Warrington

Editors-in-chief

Journal of Neuroscience Research

31st May 2021

Please consider our revised version of the manuscript entitled: "Laterodorsal tegmentum-Ventral Tegmental Area projections encode positive reinforcement signals". We would like to refer that the reviewers'

critiques were very pertinent and constructive, which contributed to improve the manuscript.

To answer all of the reviewers' concerns, we have performed substantial manuscript changes; performed and refined the analysis of electrophysiological and behavioral data; and performed additional histological experiments.

All revised text in this new version of the manuscript is marked in **red bold**, together with a clean version

To our knowledge, to date, our results still prove that LDT-VTA projections are crucial for reward value, and that its manipulation has a profound impact on behavioral output.

We believe these findings will be of interest to the readers of *Journal of Neuroscience Research* and researchers studying reward/aversion circuitry and basal ganglia, but also to a broader readership that includes

researchers studying disorders presenting reward/aversion deficits such as depression and addiction

In the name of the team, I would like to thank for the opportunity to resubmit this manuscript. We truly hope

it receives your favorable consideration in the *Journal of Neuroscience Research*. Do not hesitate to contact

us if you need any extra information.

We hope to hear from you soon.

Sincerely yours,

Ana João Rodrigues, PhD

Faculty Investigator

ICVS, School of Medicine

University of Minho

Portugal

Email: ajrodrigues@med.uminho.pt

Phone: +351 253 604 929

Laterodorsal tegmentum-Ventral Tegmental Area projections encode positive reinforcement signals

Barbara Coimbra, Ana Veronica Domingues, Carina Soares-Cunha, Raquel Correia, Luisa Pinto, Nuno Sousa, Ana Joao Rodrigues

Review timeline:

Submission date: 8th march 2021

Editorial Decision: 15th April 2021

Revision Received: 15th April 2021

Editorial Decision:

Revision Received:

Accepted:

15-Apr-2021

Dear Dr Coimbra:

Thank you for submitting your manuscript to the Journal of Neuroscience Research. We've now received the reviewer feedback and have appended those reviews below. I'm

glad to say that the reviewers are overall very enthusiastic and supportive of the study. They did raise some concerns and made some suggestions for clarification, but I expect that these points should be relatively straightforward to address. If there are any questions

or points that are problematic, please feel free to contact me. I am glad to discuss.

We ask that you return your manuscript within 30 days. Please explain in your cover letter

how you have changed the present version and submit a point-by-point response to the editors' and reviewers' comments. If you require longer than 30 days to make the revisions, please contact Dr Cristina Ghiani (cghiani@mednet.ucla.edu). To submit your revised manuscript: Log in by clicking on the link below

<https://wiley.atyponrex.com/submissionBoard/1/14c1d00e-3baa-45da-bbf0-b41c5ce5511d/current>

(If the above link space is blank, it is because you submitted your original manuscript through our old submission site. Therefore, to return your revision, please go to our new submission site here (submission.wiley.com/jnr) and submit your revision as a new manuscript; answer yes to the question "Are you returning a revision for a manuscript originally submitted to our former submission site (ScholarOne Manuscripts)? If you indicate yes, please enter your original manuscript's Manuscript ID number in the space below" and including your original submission's Manuscript ID number (jnr-2021-Mar-9632) where indicated. This will help us to link your revision to your original submission.)

The journal has adopted the "Expects Data" data sharing policy, which states that all original articles and reviews must include a Data Availability Statement (DAS). Please see <https://authorservices.wiley.com/author-resources/Journal-Authors/openaccess/data-sharing-citation/data-sharing-policy.html#standardtemplates> for examples of an appropriate DAS. Please include the DAS in the manuscript as well.

Thank you again for your submission to the Journal of Neuroscience Research; we look forward to reading your revised manuscript.

Best Wishes,

Dr Bertrand Lambolez

Associate Editor, Journal of Neuroscience Research

Dr Cristina Ghiani

Editor-in-Chief, Journal of Neuroscience Research

Authors would like to thank the editors and reviewers for the important and constructive comments which helped to clarify and improve this manuscript. In this new version, we have performed additional experiments and included reviewers' suggestions. We believe

we have addressed all of the remarks.

Editors Comments to the Author:

Please mention in the title and/or abstract that the work was done only in male rats, also discuss this as a limitation.

SEX DIFFERENCES

The National Institutes of Health now mandates the inclusion of sex as a biological variable. To conform with this mandate, the Journal of Neuroscience Research has established new policy (please see our editorial: <http://onlinelibrary.wiley.com/doi/10.1002/jnr.23979/full>) requiring all authors to ensure proper consideration of sex as a biological variable. Please ensure that: 1) Any paper utilizing subjects of one sex state the sex of the sample in the title and abstract; 2) The number of samples/subjects of each sex used in the research must be clearly stated in the methods section; 3) The inability for any reason to study sex differences where they may exist should be discussed as a study limitation. We are also encouraging authors to report exploratory analyses of potential sex differences in studies not explicitly designed to address them.

Authors' Response

The Editor raised a very important point regarding sex differences on this specific projection and thus, we clearly stated in the abstract that the study was performed in adult male rats.

We would like to clarify that in this study we did not included females, and we discussed this decision as a limitation (page 5). However, it is important to refer that in previous experiments performed in our group using the same behavioral tasks, we found no differences on the performance (considering lever pressing or reward consumption) when

comparing both sexes, as it is possible to observe in figure 1 below. Here, both males and females show no differences on the total number of lever presses during the acquisition phase of the two-choice paradigm or in the breakpoint and pellets earned during the progressive ratio task.

Figure 1 – Comparison of two-choice performance between male and female subjects submitted to the same viral approach from this study. a) Lever pressing in each session during the acquisition phase of the two-choice paradigm showed no differences between male and female subjects ($n_{\text{female}}=9$; $n_{\text{male}}=10$). b) Breakpoint of progressive ratio task for the stim+ and stim- lever and c) number

of pellets earned during each session is similar between male and females.

Reviewer 2: Comments to the Author

In this manuscript the authors assessed the role of the laterodorsal tegmentum (LDT)

1 2 3 4 5 6 7
0
100
200
300
400
500
600

Training day/Schedule

Lever presses

female YFP stim+

female YFP stim-

FR1 FR4 RR4 RR6

male YFP stim+

male YFP stimfemale-

YFP stimfemale-

YFP stim+

male-YFP stimmale-

YFP stim+

0

20

40

60

80

100

Breakpoint

Two Progressive Ratio choice reinforcement

female-YFP stimfemale-

YFP stim+

male-YFP stimmale-

YFP stim+

10

12

14

16

18

Number of pellets consumed

a

b c

projections to the ventral tegmental area (VTA) and their role on reward processing. Optical manipulations of the LDT-VTA projections are shown to influence reward value and the behavior related to it. The demonstrations have been carried out on an operant task with food reinforcers, as well as place conditioning tasks, and were accompanied by immuno-based assays showing correlation between c-fos expression in the LDT, VTA and nucleus accumbens (NAcc) and the behavioral results.

Overall, the work represents an interesting contribution on the participation of LDT on the reward system, but a few observations should be addressed:

Minor question: Page 15, line 37: The names of the figures should be corrected (Fig. 2c; line 52: Fig. 2d).

Authors' Response

We would like to thank the reviewer for pointing this out. We have now corrected figures.

Major questions:

[1] The assertion that the inhibition of the LDT-VTA projection (NpHR experiment) conveys a decrease in motivation.

As opposed to the previous experiment (with ChR2), in which animals worked more for the paired (food pellet+stimulation) reinforcer than the control group, a decreased number of stim+ paired lever presses may not necessarily convey decreased motivation in a two choice paradigm. That is: the decrease on stim+ presses (if, say, compensated by an increase in stim- presses) may rather reflect a bias in lever preference, not a change in motivational status.

It is important to show the total number of rewards obtained by each group. Despite the preference being so far convincing, a decrease in motivation would be more convincingly demonstrated by a decrease in the total number of rewards earned.

Authors' Response

We agree with the reviewer, so we provided the total number of rewards in each condition. Please see new Figure 3.

In the two-choice task, no differences were found in the total number of lever presses (stim+ and stim- combined) between YFP and ChR2 or YFP and NpHR during the acquisition phase (Sup. Fig 3a, b). However, in the PR test, ChR2 animals presented increased number of lever presses in stim+ lever in comparison to stim-; conversely, NpHR presented decreased presses in stim+ lever. In agreement, ChR2 animals earn more pellets for the stim+ lever, when compared to the number of pellets earned for the stim- lever and in comparison, to YFP animals (now Fig. 2d). Oppositely, NpHR animals earn a smaller number of pellets in the stim+ lever (now Fig. 3d). With this additional information, we now believe that the effects of the optogenetic stimulation/inhibition in motivation are clearer.

[2] Another relevant question concerns the seemingly conflicting results obtained from ChR2 experiments. Specifically, the fact that stim+ lever was subjected to extinction, even though the same stimulation induced place preference.

On this regard, the number of magazine entries will give a useful insight on potential differences in the attribution of value to the food itself, as magazine approach is a parameter sensitive to changes in the value of the outcome.

Also, the discussion should mention the behavioral and neuroanatomical substrates required for place preference and instrumental conditioning, which may give insights regarding these results.

Authors' Response

We agree with the reviewer that magazine entries would be an important measure. But unfortunately, due to technical limitations, our set up does not allow to register magazine entries.

Considering the last point raised by the reviewer, we discussed the main conceptual differences between place preference and instrumental responding that can help explain the observations (page 23).

[3] Finally, a word should be addressed to the results obtained by c-fos analyses in the LDT within the NpHR groups. That is, YFPstim+ and YFPstim- show similar patterns of c-fos expression; however, the NpHRstim- group showed a seemingly increase in c-fos (at least enough to present statistical significance when compared to the NpHRstim+ group). Given that no optical stimulation has been done, should one not expect c-fos expression in this group to be similar to the YFP groups? Authors should discuss the potential causes for this difference.

Authors' Response

For c-fos analysis, animals from all groups performed a progressive ratio test for either the stim- or the stim+ lever. After 90min, brains were collected. The increase in c-fos reported here may be explained by the performance observed during the progressive ratio. As mentioned, NpHR animals revealed a higher motivation to work for the reward associated in stim- lever, so it is plausible that the observed increase in LDT recruitment is triggered by this, rather than by a direct effect of LDT terminal stimulation (this is now discussed in page 23).

Reviewer 3: Comments to the Author

In this study, the authors investigate the impact of LDT –VTA projections on natural reward related behaviours. To do so, they successfully combine viral optogenetic approaches (excitation and inhibition of LDT-VTA inputs) and classic reward related tasks

including, progressive ratio task and place preference tests. They use electrophysiological recordings to assess the functional efficiency of the viral approaches

and use cfos staining to correlate some behavioural results with cfos neuronal activation.

While the manuscript is highly interesting, well conducted and well written and I have some suggestions that could improve the methodology and the interpretation of results.

Regarding viral approaches, the authors should demonstrate the specificity of the viral infections. According to the viral strategy used, LDT neurons that express opsins (hChR2

or NpHR) should also express WGA. In that case, in the LDT, each green cell body (YFP)

must also be red (mcherry). This is not shown or discussed in the results. Fig 1 b and c

are not enough (only YFP in LDT). The authors should also discuss the efficacy, i.e. the number or proportion of LDT neurons that express opsins.

Authors' Response

The original manuscript that described the WGA-cre construct evaluated animals 5 week

post-injection (Gradinaru et al., 2010). As mentioned in the original manuscript: "(...) we injected the previously described Cre-dependent AAV, now conditionally expressing eNpHR3.0 into motor cortex, and injected a novel WGA-Cre-expressing AAV (AAV2-EF1 α -mCherry-IRES-WGA-Cre) remotely into primary somatosensory cortex. Robust eNpHR3.0-EYFP expression was indeed observed in a distributed subset of the motor cortex neurons (Figure 2D) at 5 weeks after injection, despite the remoteness of the Cre recombinase AAV injection; in control animals without Cre recombinase, no expression is observed from these Cre-dependent AAVs (Tsai et al., 2009; Sohal et al., 2009). Consistent with the anticipated mode of trans-synaptic or transcellular transport of Cre, no mCherry-positive cell bodies were observed in motor cortex, and no EYFP-positive cell bodies were observed in S1 sensory cortex (Figure 2D)."

Consistent with this report and others (Dautan et al., 2014, 2020; Gompf et al., 2015; Gunaydin et al., 2014; Neumann et al., 2016; Sanders & Jaeger, 2016) and our previous

studies using the same viral approach (Coimbra et al., 2017, 2019), we did not find mCherry-positive cells in the LDT as anticipated, considering the characteristics of this virus.

Regarding the number or proportion of LDT neurons that express opsins, we now included this data in the manuscript in Fig. 1d-e. The reviewer can observe that ~36.5% of LDT neurons express ChR2 or NpHR opsins.

Regarding electrophysiological approaches, some important information is missing. For the optotagging approach, the authors write, "Neurons showing a firing rate increase or decrease by more than 20% from the mean frequency of the baseline period were considered as responsive."

- What is the duration of the baseline period?

Authors' Response

We reported the duration of the baseline period in the Methods section of the manuscript.

The spontaneous activity for the baseline was recorded for 60s per cell.

- Why use 20% increase or decrease in activity?

Authors' Response

For the purpose of identifying neurons that increased, decreased or did not change activity from baseline, we defined a criteria of change of firing rate 20% above or below average activity of the baseline, as previously reported by others and us (Benazzouz et al., 2000; Coimbra et al., 2017, 2019; Soares-Cunha et al., 2016, 2019). This information

was now added to the methods section of the manuscript.

- What are the responses to each 10ms pulse? Do the neurons follow 80 stimulations at 20hz? The traces in fig 1 g and I are not sufficient. The authors should use more appropriate time resolution to respond to above questions.

Authors' Response

We agree that this information requested by the reviewer is important to include. We altered the graphs in Fig. 1 (now Fig. 1i and Fig.1l) in order to depict a higher time resolution (0.05s bin size). As expected, not all cells respond to each pulse, however, it is possible to observe that activation of LDT inputs onto dopaminergic neurons induces a

significant increase in the firing rate of these cells and a decrease in GABAergic inhibition

of LDT inputs showed an opposite effect. In addition, we present a heatmap of percentages of firing rate change from baseline, in order to better represent peak and offset dynamics of activation/inhibition of LDT terminals (Fig. 1j, m and page 15).

- Do panel's g and l in figure 1 represent a single stimulation period (80 stims) in one animal, or several stimulation periods in several animals? In the second case, how the authors did the normalization. Error bars are also missing if these traces correspond to an average. What those traces represent is unclear.

Authors' Response

We would like to thank the reviewer for this pertinent point. In fact, panels in Fig.1 represent the mean of all dopaminergic or GABAergic cells recorded from all animals (ChR2 or NpHR groups). In each animal, several cells are recorded in the VTA region (recording electrode lowered in increments of 100 μ m, dorsoventral ranging from -7.5 to -8.2). Every cell is recorded for a baseline of 60s, followed by activation/inhibition stimulus

(~4s) and recorded for an additional 60s; only one trial per cell is performed. The data is not normalized, but rather an average of the recorded firing rate of all cells for each cell type. Here, we depicted the traces as error (SEM) for the shading and the mean as the full line (now clarified in the manuscript and depicted a different time resolution).

- In the results section, the authors mentioned that "Activation of LDT terminals evoked excitatory responses in 79% (55 out of 70 cells) of putative VTA dopaminergic (pDAergic) neurons and inhibitory responses in 71% (10 cells out of 14) putative VTA GABAergic (pGABAergic) neurons recorded. These numbers are high. Are they supported by literature? The similar question should be answered for optical inhibition (large proportion of responding VTA neurons).

Authors' Response

We would like to thank the reviewer for this commentary. In the table below you can find some studies with the % of response, though the electrophysiological methodology was not similar to our study. Still, % of response are very heterogenous.

Method Stimulation

%of DA response %non-DA response Reference

Slice electrophysiology of ChAT-cre rats Photo-excitation of LDT ChAT terminals in VTA Around 30% response of total VTA cells (Xiao et al., 2016) Slice electrophysiology of B16 mice using retrobeads and AAV-ChR2 Photo-excitation of LDT terminals in VTA Does not show overall percentage of response. However, it states that LDT inputs generated EPSCs in 100% of dopamine neurons projecting to the NAc lateral shell but only in 30–40% projecting to the NAc medial shell or in the substantia nigra; only 10% of dopamine neurons projecting to the mPFC yielded EPS Cs. (Lammel et al., 2012) Juxtacellular recordings in vivo of ChAT-cre rats Photo-excitation of LDT ChAT terminals in VTA Excited (E, 50%), nonresponsive (NR, 42%) and inhibited (I, 8%) (clusterbased permutation test, 200 permutations). E, 35%; NR, 41%; I, 24% (Dautan et al., 2016) Juxtacellular

recordings in vivo of rats Electrical stimulation of the LDT No percentages shown, LDT stimulation elicits burst firing in putative VTA DA neurons (Lodge & Grace, 2006). Juxtacellular recordings in vivo of mice Chemogenetic inhibition of the LDT LDT inhibition decreased around 40% discharge frequency and 80% of bursting activity of VTA DA neurons

(Fernandez et al., 2018)

-p. 15 line 24 “To test the impact of LDT-VTA manipulation in behavior, we unilaterally activated these inputs in freely moving rats during a two-choice instrumental task (Fig. 2a)” fig 2a is a schematic of the viral /optogenetic stimulation procedure. The behavioral approach seems to be represented in fig 2b. There is a gap between the main text and the panels in figure 2. The letter does not match the graphic. The same is present in the legend of the figure.

Authors’ Response

We have corrected this in the manuscript.

-p. 16 lines 44-45 “Similarly, two-way ANOVA analysis showed a significant difference in the interaction between groups and chamber side (Fig. 2j; two-way ANOVA, $F(1, 20) = 26.93$, $p < 0.001$).” No corresponding graph in the figure 2

Authors’ Response

We have corrected this in the manuscript.

-p. 17 line 14. Nac instead of VTA?

Authors’ Response

We noticed this mistake and have corrected this in the manuscript.

- p. 17 and figure 3. Same as for figure 2: the drawing / graph represented in the figure do not correspond to the description in the main text. The lettering does not match the graphs.

Authors’ Response

We have corrected this throughout the manuscript.

-p18line 1-10. Supplemental fig 2 corresponds to the experimental design of behavioural

studies and not to the activation pattern of neuronal populations after the PR tasks as described in the main text. Is that the right figures? Is this the right figure? If so, the authors should adapt the description of the figure in the main text or the title of the legend.

The description of the different panels of the figure is missing in the main text.

Authors’ Response

In the revised version of the manuscript, we have clarified and corrected these points. Indeed, Supplemental Figure 2 refers to the experimental design of electrophysiological and behavioral experiments, as mentioned in the methods section of the manuscript. Supplemental Figure 2c refers to experimental design for c-fos induction. Here, animals were divided to perform a PR session for either stim+ or stim- lever and were sacrificed 90 min after the initiation of the task (mentioned in the methods section and now page 19). The activation pattern of neuronal populations after the PR task is depicted as c-fos cell density in Supplemental Figure 4.

References

- Benazzouz, A., Gao, D. M., Ni, Z. G., Piallat, B., Bouali-Benazzouz, R., & Benabid, A. L. (2000). Effect of high-frequency stimulation of the subthalamic nucleus on the neuronal activities of the substantia nigra pars reticulata and ventrolateral nucleus of the thalamus in the rat. *Neuroscience*, *99*(2), 289–295.
- Coimbra, B., Soares-Cunha, C., Borges, S., Vasconcelos, N. A., Sousa, N., & Rodrigues, A. J. (2017). Impairments in laterodorsal tegmentum to VTA projections underlie glucocorticoid-triggered reward deficits. *eLife*, *6*. <https://doi.org/10.7554/eLife.25843>
- Coimbra, B., Soares-Cunha, C., Vasconcelos, N. A. P., Domingues, A. V., Borges, S., Sousa, N., & Rodrigues, A. J. (2019). Role of laterodorsal tegmentum projections to nucleus accumbens in reward-related behaviors. *Nature Communications*, *10*(1), 4138. <https://doi.org/10.1038/s41467-019-11557-3>
- Dautan, D., Huerta-Ocampo, I., Gut, N. K., Valencia, M., Kondabolu, K., Kim, Y., Gerdjikov, T. V., & Mena-Segovia, J. (2020). Cholinergic midbrain afferents modulate striatal circuits and shape encoding of action strategies. *Nature Communications*, *11*. <https://doi.org/10.1038/s41467-020-15514-3>
- Dautan, D., Huerta-Ocampo, I., Witten, I. B., Deisseroth, K., Bolam, J. P., Gerdjikov, T., & Mena-Segovia, J. (2014). A Major External Source of Cholinergic Innervation of the Striatum and Nucleus Accumbens Originates in the Brainstem. *The Journal of Neuroscience*, *34*(13), 4509–4518. <https://doi.org/10.1523/JNEUROSCI.5071-13.2014>
- Dautan, D., Souza, A. S., Huerta-Ocampo, I., Valencia, M., Assous, M., Witten, I. B., Deisseroth, K., Tepper, J. M., Bolam, J. P., Gerdjikov, T. V., & Mena-Segovia, J. (2016). Segregated cholinergic transmission modulates dopamine neurons integrated in distinct functional circuits. *Nature Neuroscience*, *19*(8), 1025–1033. <https://doi.org/10.1038/nn.4335>
- Fernandez, S. P., Broussot, L., Marti, F., Contesse, T., Mouska, X., Soiza-Reilly, M., Marie, H., Faure, P., & Barik, J. (2018). Mesopontine cholinergic inputs to midbrain dopamine neurons drive stress-induced depressive-like behaviors. *Nature Communications*, *9*. <https://doi.org/10.1038/s41467-018-06809-7>
- Gompf, H. S., Budygin, E. A., Fuller, P. M., & Bass, C. E. (2015). Targeted genetic manipulations of neuronal subtypes using promoter-specific combinatorial AAVs in wild-type animals. *Frontiers in Behavioral Neuroscience*, *9*. <https://doi.org/10.3389/fnbeh.2015.00152>
- Gradinaru, V., Zhang, F., Ramakrishnan, C., Mattis, J., Prakash, R., Diester, I., Goshen, I., Thompson, K. R., & Deisseroth, K. (2010). Molecular and Cellular Approaches for Diversifying and Extending Optogenetics. *Cell*, *141*(1), 154–165. <https://doi.org/10.1016/j.cell.2010.02.037>
- Gunaydin, L. A., Grosenick, L., Finkelstein, J. C., Kauvar, I. V., Fenno, L. E., Adhikari, A., Lammel, S., Mirzabekov, J. J., Airan, R. D., Zalocusky, K. A., Tye, K. M., Anikeeva, P., Malenka, R. C., & Deisseroth, K. (2014). Natural neural projection dynamics underlying social behavior. *Cell*, *157*(7), 1535–1551. <https://doi.org/10.1016/j.cell.2014.05.017>
- Lammel, S., Lim, B. K., Ran, C., Huang, K. W., Betley, M. J., Tye, K. M., Deisseroth, K., & Malenka, R. C. (2012). Input-specific control of reward and aversion in the ventral tegmental area. *Nature*, *491*(7423), 212–217. <https://doi.org/10.1038/nature11527>
- Lodge, D. J., & Grace, A. A. (2006). The laterodorsal tegmentum is essential for burst firing of ventral tegmental area dopamine neurons. *Proceedings of the National Academy of Sciences of the United States of America*, *103*(13), 5167–5172. <https://doi.org/10.1073/pnas.0510715103>
- Neumann, P. A., Wang, Y., Yan, Y., Wang, Y., Ishikawa, M., Cui, R., Huang, Y. H., Sesack, S. R., Schlüter, O. M., & Dong, Y. (2016). Cocaine-Induced Synaptic Alterations in Thalamus to Nucleus Accumbens Projection. *Neuropsychopharmacology*, *41*(9), 2399–2410. <https://doi.org/10.1038/npp.2016.52>
- Sanders, T. H., & Jaeger, D. (2016). OPTOGENETIC STIMULATION OF CORTICO-SUBTHALAMIC PROJECTIONS IS SUFFICIENT TO AMELIORATE BRADYKINESIA IN 6-OHDA LESIONED MICE. *Neurobiology of Disease*, *95*, 225–237. <https://doi.org/10.1016/j.nbd.2016.07.021>
- Soares-Cunha, C., Coimbra, B., David-Pereira, A., Borges, S., Pinto, L., Costa, P., Sousa, N., & Rodrigues, A. J. (2016). Activation of D2 dopamine receptor-expressing neurons in the nucleus accumbens increases motivation. *Nature Communications*, *7*, 11829. <https://doi.org/10.1038/ncomms11829>

Soares-Cunha, C., de Vasconcelos, N. A. P., Coimbra, B., Domingues, A. V., Silva, J. M., Loureiro-Campos, E., Gaspar, R., Sotiropoulos, I., Sousa, N., & Rodrigues, A. J. (2019). Nucleus accumbens medium spiny neurons subtypes signal both reward and aversion. *Molecular Psychiatry*. <https://doi.org/10.1038/s41380-019-0484-3>

Xiao, C., Cho, J. R., Zhou, C., Treweek, J. B., Chan, K., McKinney, S. L., Yang, B., & Gradinaru, V. (2016). Cholinergic Mesopontine Signals Govern Locomotion and Reward through Dissociable Midbrain Pathways. *Neuron*, 90(2), 333–347. <https://doi.org/10.1016/j.neuron.2016.03.028>

2nd Editorial Decision

Decision Letter

Dear Dr Coimbra:

Thank you for resubmitting your manuscript.

You will be pleased to know that your manuscript has been accepted for publication. Thank you for submitting this excellent work to our journal.

In the coming weeks, the Production Department will contact you regarding a copyright transfer agreement and they will then send an electronic proof file of your article to you for your review and approval.

Please note that your article cannot be published until the publisher has received the appropriate signed license agreement. Within the next few days, the corresponding author will receive an email from Wiley's Author Services asking them to log in. There, they will be presented with the appropriate license for completion. Additional information can be found at <https://authorservices.wiley.com/author-resources/Journal-Authors/licensing-open-access/index.html>

Would you be interested in publishing your proven experimental method as a detailed step-by-step protocol? Current Protocols in Neuroscience welcomes proposals from prospective authors to disseminate their experimental methodology in the rapidly evolving field of neuroscience. Please submit your proposal here: <https://currentprotocols.onlinelibrary.wiley.com/hub/submitproposal>

Congratulations on your results, and thank you for choosing the Journal of Neuroscience Research for publishing your work. I hope you will consider us for the publication of your future manuscripts.

Sincerely,

Dr Bertrand Lambolez
Associate Editor, Journal of Neuroscience Research

Dr Cristina Ghiani
Editor-in-Chief, Journal of Neuroscience Research

Associate Editor: Lambolez, Bertrand
Comments to the Author:
(There are no comments.)

Reviewer: 2

Comments to the Author
The authors have adequately addressed the points raised and the manuscript should be considered for publication.

Reviewer: 3

Comments to the Author
The authors answered all my questions and greatly improved the manuscript. I have no further concerns about the manuscript. In my opinion, the manuscript can be published.

Authors' Response

3rd Editorial Decision

Decision Letter

Authors' Response

4th editorial decision

Decision Letter

Author response